# Structural insight into the activation mechanism of MrgD with heterotrimeric Gi-protein revealed by cryo-EM

Shota Suzuki[1,8], Momoko Iida[2], Yoko Hiroaki [3,4], Kotaro Tanaka[1,3], Akihiro Kawamoto [5,6], Takayuki Kato [5] & Atsunori Oshima [1,3,7✉]

MrgD, a member of the Mas-related G protein-coupled receptor (MRGPR) family, has high basal activity for Gi activation. It recognizes endogenous ligands, such as β-alanine, and is involved in pain and itch signaling. The lack of a high-resolution structure for MrgD hinders our understanding of whether its activation is ligand-dependent or constitutive. Here, we report two cryo-EM structures of the MrgD-Gi complex in the β-alanine-bound and apo states at 3.1 Å and 2.8 Å resolution, respectively. These structures show that β-alanine is bound to a shallow pocket at the extracellular domains. The extracellular half of the sixth transmembrane helix undergoes a significant movement and is tightly packed into the third transmembrane helix through hydrophobic residues, creating the active form. Our structures demonstrate a structural basis for the characteristic ligand recognition of MrgD. These findings provide a framework to guide drug designs targeting the MrgD receptor.

[1] Department of Basic Medicinal Sciences, Graduate School of Pharmaceutical Sciences, Nagoya University, Furo-cho, Chikusa-ku, Nagoya 464-8601, Japan. [2] Division of Biological Science, School of Science, Nagoya University, Furo-cho, Chikusa-ku, Nagoya 464-8601, Japan. [3] Cellular and Structural Physiology Institute (CeSPI), Nagoya University, Furo-cho, Chikusa-ku, Nagoya 464-8601, Japan. [4] Japan Biological Informatics Consortium (JBIC), Nagoya University, Furo-cho, Chikusa-ku, Nagoya 464-8601, Japan. [5] Institute for Protein Research, Osaka University, Suita, Osaka 565-0871, Japan. [6] Japan Science and Technology Agency, PRESTO, Saitama 332-0012, Japan. [7] Institute for Glyco-core Research (iGCORE), Nagoya University, Furo-cho, Chikusa-ku, Nagoya 464-8601, Japan. [8] Present address: Advanced Research Institute, Tokyo Medical and Dental University, 1-5-45 Yushima, Bunkyo-ku, Tokyo 113-8510, Japan. ✉email: atsu@cespi.nagoya-u.ac.jp

A family of GPCRs named MAS-related G protein-coupled receptors (MRGPRs) was discovered 20 years ago[1–3]. The MRGPR family consists of ~40 members grouped into nine distinct subfamilies (A–H and X). The physiological role of the MRGPR family is still unclear, but some evidence supports a role for MRGPR subtypes in nociception, pruritus, cell proliferation, circulation, and mast cell degranulation[3,4]. The MRGPR family members have been considered orphan receptors. However, for several receptors, their physiological ligands have been recently identified[5–8].

Mas-related-G protein-coupled receptor D (MrgD) is expressed in sensory dorsal root ganglia (DRG) neurons, the cardiovascular system, and the retina[7,9–11]. A variety of endogenous agonists, including β-alanine[7,12,13], alamandine[14], and angiotensin (1–7) (Ang (1–7))[15] have been reported. β-alanine is mainly involved in the pain and itch signaling of MrgD[12]. For example, β-alanine activates Gi and Gq via MrgD in DRG neurons, followed by inactivation of the KCNQ channels and increased neuronal excitability[16]. β-alanine also activates TRPA channels through Gs-activation and through PKA-activation, which is essential for cold allodynia in chronic constriction injury-induced neuropathic pain[17]. The MrgD is also activated by 5-oxoeicosatetraenoic acid, a metabolite of polyunsaturated fatty acid. This pathway induces somatic and visceral pain without inflammation and causes unpleasant symptoms in patients with constipated irritable bowel syndrome[18]. All these pathways are involved in regulating neuronal excitability; however, which pathway is utilized under physiological conditions remains unclear.

MrgD also recognizes Ang (1–7) and alamandine, which are metabolites of angiotensin II (AngII), and antagonizes activation of angiotensin II type 1 receptor (AT1R) by AngII in the renin–angiotensin system pathway to relax blood vessels and lower blood pressure[10,14,19–21]. In rats, alamandine-activated MrgD-induced nitric oxides release, suggesting that MrgD may be involved in cardioprotection through the regulation of cardiac vasodilation and fibrosis[22,23]. Genetic ablation of MrgD in mice reduced the mechanical nociceptive capacity of sensory neurons and caused dilated cardiomyopathy[24,25]. These activities of MrgD make it an attractive drug target for the regulation of blood pressure, neuropathic pain, depression, retinal diseases, and myocardial health.

Some GPCRs, including MrgD, have also been reported to have high basal activity. For example, HEK293 cells overexpressing MrgD showed constitutive activity through Gi and Gq[13]. The creation of stable cell lines expressing MrgD has also revealed that some clones with high transcription levels become unresponsive to β-alanine[26]. Recent studies have shown that HeLa cells expressing MrgD show a ligand-independent release of IL-6 through Gq signaling[27]. Although many structures of various GPCR-G protein complexes have been reported, little structural information is available for the ligand-free (apo) state, leaving the mechanism for this ligand-independent activation unclear.

The lack of a high-resolution structure for MrgD has hindered the determination of its endogenous ligand recognition and activation mechanisms. MrgD, like all MRGPR family members, is expected to have a unique activation mechanism due to its lack of the conserved toggle switch (W6.48) motif[28,29] normally required for GPCR activation.

This study used single-particle cryo-electron microscopy (cryo-EM) to determine the two states of the MrgD-Gi complex: one is the endogenous agonist β-alanine-bound state, and the other is the ligand-free (apo) state. We found that β-alanine binds to a shallow extracellular pocket of MrgD. In addition, TM6 of MrgD significantly tilts toward the TM3 side upon ligand-binding, and a tight interaction of the bulky residues occurs between TM3 and TM6, leading to stabilizing the active conformation. These findings demonstrate molecular recognition of β-alanine-induced and constitutive activation of MrgD.

## Results

**Cryo-EM analysis of the MrgD–Gi complexes**. We generated an MrgD construct with thermostabilized bRIL[30] conjugated at the N-terminus of the receptor. Deletion of four amino acids in the N-terminus of MrgD improved receptor expression level and stability. The modified MrgD exhibited Gi-coupling activity comparable to that of the wild-type (WT), as measured by a NanoBiT-G-protein dissociation assay[31] (Supplementary Fig. 1a, b, and Table 1). We used dominant-negative (DN) Gi for the structural determination, as this has been reported to increase the stability of the complex[32,33]. The MrgD receptor was co-expressed with DNGi and Gβγ to generate the complex. The membrane was co-incubated with the endogenous agonist β-alanine, along with apyrase to remove guanine nucleotides. In the case of the apo state, no ligand was added during all the purification steps. After FLAG-affinity purification, scFv16, which binds to the interface of the G protein α- and β-subunits, was added. This procedure allowed the stable assembly of an MrgD DNGi-Gβγ (MrgD–Gi) complex (Supplementary Figs. 2–4).

The high-resolution structures of the β-alanine-bound and apo MrgD–Gi complexes were determined using cryo-EM at an overall resolution of 3.1 and 2.8 Å, respectively (Fig. 1, Supplementary Figs. 2e, 3e, Table 1). In the complex structures, the local resolution was highest for the Gβ and Gα subunits stabilized by scFv16 and lowest for the extracellular surface of the receptor (Supplementary Figs. 2e and 3e). The α-helical domain of Gα was not resolved due to its flexibility. These structures show a canonical GPCR fold of seven transmembranes (TM) segments surrounded by an annular detergent micelle mimicking the natural phospholipid bilayer (Fig. 1). As is typical of many GPCR–G protein complex structures, our initial maps yielded poor resolution for β-alanine and extracellular loops (ECLs). We improved the density using local refinement covering only the TM domain on cryoSPARC. The improved maps enabled the construction of an atomic model for β-alanine and the ECLs (Supplementary Figs. 2f, 3f, and 5).

**Ligand-binding pocket of the MrgD receptor**. The overall structure of the β-alanine-bound state is similar to that of the apo state, with a root-mean-square deviation (RMSD) of 0.544 Å for the Cα atoms. The ligand-binding pocket of the cryo-EM structure with β-alanine shows a significant density on the extracellular surface side; therefore, β-alanine was assigned to it (Fig. 2a, b). We determined the orientation of β-alanine based on the electrostatic potential of the ligand-binding pocket surface (Fig. 2c). The β-alanine forms hydrogen bonds with the surrounding polar residues of TM3 and TM5. The β-alanine carboxyl group interacts with R103$^{3.30}$ of TM3, and the amino group on the opposite side is bound to D179$^{5.37}$ of TM5 and is sufficiently close to form an H-bond with the backbone of W241$^{6.55}$ and the carboxyl group of the ligand with the backbone of C164$^{4.64}$ (Fig. 2d, e). The β-alanine is also surrounded by the hydrophobic residues C164$^{4.64}$, C175$^{5.33}$, W241$^{6.55}$, Y245$^{6.59}$, and W246$^{6.60}$ (Fig. 2d, e, Supplementary Fig. 6).

To investigate the contribution of these residues to ligand binding, we measured β-alanine-dependent Gi signaling activity of point mutants using the NanoBiT-G-protein dissociation assay[31]. The signaling activity was eliminated by R103A, D179A, W241A, and W246A and decreased by Y245A (Fig. 2f, Supplementary Fig. 7, Table 2, Supplementary Data 1). To evaluate the binding orientation of β-alanine, we performed MD

**Table 1 Cryo-EM data collection, refinement, and validation statistics.**

| | β-alanine-bound MrgD–Gi (EMDB- 33554) (PDB 7Y12) | apo MrgD–Gi (EMDB-33557) (PDB 7Y15) | β-alanine-bound MrgD, local (EMDB- 33556) (PDB 7Y14) | apo MrgD, local (EMDB-33555) (PDB 7Y13) |
|---|---|---|---|---|
| *Data collection and processing* | | | | |
| Magnification | 105,000 | 105,000 | | |
| Voltage (kV) | 300 | 300 | | |
| Electron exposure (e–/Å$^2$) | 60 | 60 | | |
| Defocus range (μm) | −0.5 to −1.7 | −0.5 to −1.7 | | |
| Pixel size (Å) | 0.675 | 0.675 | | |
| Symmetry imposed | C1 | C1 | | |
| Final particle images (no.) | 97,282 | 349,331 | | |
| Map resolution (Å) | 3.1 | 2.8 | 3.2 | 3.0 |
| FSC threshold | 0.143 | 0.143 | 0.143 | 0.143 |
| Map resolution range (Å) | 2.8–4.8 | 2.8–4.8 | 2.8–4.8 | 2.8–4.8 |
| *Refinement* | | | | |
| Initial model used (PDB code) | Swiss model (AT1R as a template), 6N4B | β-alanine-bound MrgD–Gi | | |
| Model resolution (Å) | 3.2 | 3.1 | 3.4 | 3.2 |
| FSC threshold | 0.5 | 0.5 | 0.5 | 0.5 |
| *Model composition* | | | | |
| Non-hydrogen atoms | 8.507 | 8495 | 2144 | 2198 |
| Protein residues | 1109 | 1109 | 265 | 264 |
| Ligands | | | | |
| *R.m.s. deviations* | | | | |
| Bond lengths (Å) | 0.003 | 0.002 | 0.004 | 0.005 |
| Bond angles (°) | 0.56 | 0.509 | 0.870 | 0.919 |
| *Validation* | | | | |
| MolProbity score | 1.58 | 1.82 | 1.76 | 1.85 |
| Clashscore | 8.07 | 7.85 | 8.67 | 8.17 |
| Poor rotamers (%) | 0 | 0 | 0 | 0 |
| *Ramachandran plot* | | | | |
| Favored (%) | 97.3 | 97.71 | 95.8 | 93.9 |
| Allowed (%) | 2.7 | 2.29 | 4.2 | 6.1 |
| Disallowed (%) | 0 | 0 | 0 | 0 |

simulations. β-alanine remains bound at this site stably throughout the 1 μs MD simulations, as assessed by the small fluctuations in the RMSD and the distances to the surrounding residues (Supplementary Fig. 8). Further, we generated the model that has β-alanine in the opposite orientation and started the simulation in three independent runs. The orientation of β-alanine was reversed in the equilibration steps in all runs, indicating that our modeling of β-alanine is reasonable (Supplementary Fig. 9). These analyses suggest that R103$^{3.30}$ and D179$^{5.37}$ are required for binding to β-alanine, whereas W241$^{6.55}$, Y245$^{6.59}$, and W246$^{6.60}$ play crucial roles in forming the ligand-binding pocket and activation of the receptor.

The structure of the β-alanine-bound state shows that the ligand-binding pocket of MrgD is shallow and that the bound β-alanine is exposed to the extracellular solvents (Fig. 2a, c). The binding site is near ECL2, away from S$^{6.48}$, which is characteristic of MrgD (Supplementary Fig. 10). In MrgD, a disulfide bond is formed between C164$^{4.64}$ in TM4 and C175$^{5.33}$ in TM5, which is not observed in other class A GPCRs (Fig. 2d, Supplementary Fig. 11a). The NanoBiT-G-protein dissociation assay showed that the C164S and C175S mutations reduced the Gi signaling (Supplementary Fig. 11b, Supplementary Data 1). The two cysteines are conserved in the MRGPR family (Supplementary Fig. 11c). The expression level of these two mutants (C164S, C175S) on the cell surface was markedly reduced (Supplementary Fig. 7). These results indicate an essential contribution of the disulfide bond to proper folding and trafficking.

**MrgD–Gi complex in the active state.** Since the overall structures of the two MrgD–Gi complexes resembled each other, we used the β-alanine-bound form as a representative for conformational assessment, unless otherwise noted. The MrgD–Gi complexes adopt active conformations similar to the closely related active AT1R (The RMSD with AT1R is 1.07 for the Cα atoms, the sequence identity is 14%, and the sequence similarity is 34%) (Fig. 3a). The structural alignment of the MrgD receptor with inactive (PDB code: 4YAY)[34] and active (PDB code: 6OS0)[35] states of AT1R revealed the basis for the activation of MrgD. Compared to the inactive AT1R, the Cα atom of P216$^{6.30}$ at the TM6 cytoplasmic edge of MrgD is shifted outward by 11.7 Å (Fig. 3a). The TM7 cytoplasmic end at the Cα atom of Y276$^{7.53}$ is moved inward by 5.0 Å (Fig. 3a). The outward shift of TM6 is a hallmark of class A GPCR activation, which allows the C-terminal α5 helix of the Gα subunit to bind to the receptor core, initiating signal transduction.

Class A GPCRs have several conserved motifs, such as CWxP, PIF, DRY, and NPxxY, associated with receptor activation[36]. In MrgD, the P$^{5.50}$I$^{3.40}$F$^{6.44}$ motif is replaced by the sequence of P192$^{5.50}$, L113$^{3.40}$, and F230$^{6.44}$. The rotameric orientation of the side chains L113$^{3.40}$ and F230$^{6.44}$ is consistent with active AT1R, but the side chain of P192$^{5.50}$ in MrgD is oriented 180° opposite P207$^{5.50}$ of the active AT1R (Fig. 3b). Instead of P192$^{5.50}$, L190$^{5.48}$ forms a hydrophobic interaction with L113$^{3.40}$ and F230$^{6.44}$ in MrgD (Fig. 3b), and the kink angle of TM5 is reversed from AT1R (Fig. 3b). Eventually, the extracellular side of TM5 shifts toward TM4 in the MrgD (Supplementary Fig. 11a). The highly conserved D$^{3.49}$R$^{3.50}$Y$^{3.51}$ (DRY) motif and N$^{7.49}$P$^{7.50}$Y$^{7.53}$ (NPxxY) motif are present near the intracellular crevice in class A GPCRs. The sequence corresponding to the DRY motif of MrgD comprises Q122$^{3.49}$, R123$^{3.50}$, and C124$^{3.51}$.

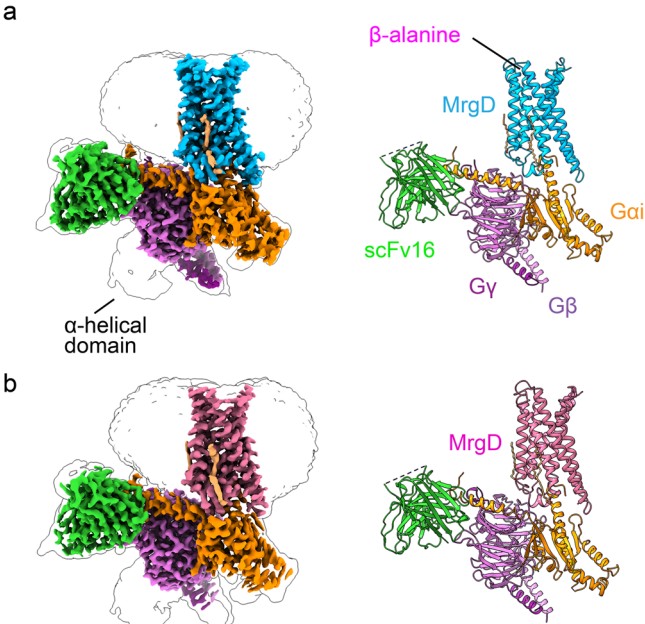

**Fig. 1 Overall structure of the MrgD–Gi complex. a** β-alanine-bound state of the MrgD–Gi complex. **b** apo state of the MrgD–Gi complex. The orthogonal views of the cryo-EM density map (left) and the built ribbon models (right) are shown. β-alanine-bound MrgD is in cyan, and Apo-MrgD is colored in red. β-alanine is shown in magenta, and palmitic acid is light yellow. The Gi heterotrimer is color-coded by subunits: Gi is in orange, Gβ in orchid, Gγ2 in purple, and scFv16 in green. The color scheme of the model is consistent with the density map.

The side chain of $R123^{3.50}$ extends toward TM7 (Fig. 3c). $Y276^{7.53}$ of the NPxxY motif on TM7 of MrgD is displaced 5 Å toward the center of the cavity, compared to the inactive AT1R (Fig. 3d). Collectively, MrgD lacks a canonical PIF motif but involves a cascade of conformational changes through rearrangement of D(Q) RY (C) and NPxxY motifs characterized in most class A GPCRs.

**Characteristic structural rearrangement of TM6 stabilizes the activation state of MrgD.** We characterized the distinct conformational feature of MrgD by comparison with three representative class A GPCRs, namely an active form of AT1R[35], dopamine receptor type 3 (D3R)(PDB code: 7CMU)[37], and cannabinoid receptor type 1 (CB1) (PDB code: 6N4B)[38]. The most characteristic conformational change of MrgD is found in the TM3 and TM6 regions. The extracellular half of TM6 is tilted about 20° toward TM3, while the cytoplasmic half of TM6 is in good agreement with other active state GPCR structures (Fig. 4a). The result is a narrower distance between the extracellular half of TM3 and TM6 in MrgD than in the other three receptors (Supplementary Fig. 12a–d). $Y245^{6.59}$ and $W241^{6.55}$ are directly involved in the binding of β-alanine, and $W241^{6.55}$ and $Y106^{3.33}$ undergo π–π stacking (Fig. 4b). $Y106^{3.33}$ forms hydrophobic interactions with $L237^{6.51}$ and $Y109^{3.36}$. Consequently, TM3 and TM6 are tightly packed together. In AT1R, D3R, and CB1, the deep ligand-binding site is formed by the side chains of TM3, TM5, TM6, and ECL2, so no interaction occurs between the side chains of the extracellular side of TM3 and TM6 (Supplementary Fig. 12a–d).

Most class A GPCRs have a conserved $C^{6.47}W^{6.48}xP^{6.50}$ motif, known as a toggle switch, but MrgD has Serine at position 6.48. In GPCRs with the canonical toggle switch, a deeply bound ligand pushes $W^{6.48}$ directly and triggers the rotation of the side chains

of $F^{6.44}$ and $I^{3.40}$ in the PIF motif[36]. By contrast, the $S234^{6.48}$ of MrgD is far from the binding position of β-alanine and cannot interact directly with the ligand (Fig. 4b, Supplementary Fig. 12). The binding of β-alanine triggers the formation of π–π stacking between $W241^{6.55}$ and $Y106^{3.33}$, stabilizing the TM3 interaction of TM6. Position 3.36 of TM3 is also known to regulate GPCR activation[39]. The $Y109^{3.36}$ faces TM6 and forms a hydrogen bond with the backbone carbonyl of $S234^{6.48}$, which stabilizes the active form. The side chains of $Y109^{3.36}$ and $S234^{6.48}$ in MrgD tilt toward the cytoplasmic side (Supplementary Fig. 12a). This is in contrast to the other class A GPCRs in which they are oriented to the extracellular side (Supplementary Fig. 12b–d).

Further, we measured the Gi signaling activity of the S234A mutant. S234A did not affect potency but reduced efficacy (Fig. 4c, Supplementary Fig. 7, Table 2), indicating that the serine residue at this position could be involved in a movement of the extracellular half of TM6 toward TM3.

Our structure demonstrates that the tight packing of the bulky residues in the extracellular halves between TM3 and TM6 is essential to activating MrgD. To confirm it functionally, we performed NanoBiT–G-protein dissociation assays on the alanine and phenylalanine mutants of $Y106^{3.33}$, $Y109^{3.36}$, $L237^{6.51}$, $W241^{6.55}$, and $F242^{6.56}$ (Fig. 4c). The Y106A, Y109A, L237A mutants reduced Gi signaling activity, while W241A and F242A mutants abolished the activity (Fig. 4c, Table 2, Supplementary Data 1). The Y106F, Y109F, and W241F mutants exhibited higher efficacy and potency than the alanine mutant, although the activity was slightly lower than that of WT (Fig. 4c). The Y106A/Y109A mutants showed comparable expression levels to the WT, but Gi signaling activity was abolished (Fig. 4c, Supplementary Fig. 6, Supplementary Data 1). These results support an essential function for the bulky hydrophobic residues in the extracellular half of TM3 and TM6 in the β-alanine-dependent activation of MrgD.

**MrgD–Gi interface.** The interactions between MrgD and Gi are similar to those seen in other Gi-bound GPCR complexes. The C-terminal half of the α5 helix of Gi inserts into the cavity at the cytoplasmic region of MrgD. The hydrophobic surface of the C-terminal α5 helix formed by L344, L348, and L353 interacts with the hydrophobic patches, including $V127^{3.54}$ in TM3, and $L201^{5.59}$ and $V205^{5.63}$ in TM5 and $L219^{6.33}$ and $V223^{6.37}$ in TM6 (Fig. 5a). The hydrophilic interactions, including $R282^{8.49}$–N346, $S126^{3.53}$–N347, $R282^{8.49}$–D350, $R123^{3.50}$–C351, and $R218^{6.32}$–G352, further mediate the association of MrgD and Gi (Fig. 5b). $R123^{3.50}$ of TM3 interacts with the main chain of C351, which is often found in GPCRs[37,40,41]. $R^{3.50}$ is involved in the guanine nucleotide exchange factor (GEF) activity of GPCRs, as mutations of $R^{3.50}$ have been reported to reduce GDP/GTP exchange[42–44].

The intracellular loop 2 (ICL2) of MrgD forms a short helix commonly found in GPCRs in the active state. ICL2 also undergoes hydrophobic interactions with Gi, and the side chain of $I131^{34.51}$ is packed in the αN-α5 cleft formed by L194, F366, and I343 (Fig. 5c). The hydrophobic residue at 34.51 is highly conserved in Gi-coupled receptors[37,38,40,41,45,46], and this is also involved in the interaction of MrgD and Gi (Supplementary Fig. 13a–g). The MrgD–Gi complex is stabilized by hydrogen bonds contributed by the backbone carbonyls of A31, $R137^{34.57}$, and $C135^{34.55}$ and the side chains of $K134^{34.54}$ and R32 (Fig. 5d, Supplementary Fig. 13a). The interactions between ICL2 and Gi are not common in other Gi-bound GPCR complexes (Supplementary Fig. 13b–g). When the receptor portion of each complex is superimposed, the orientation of the αN helix of Gi is diverse (Supplementary Fig. 13h). The different interactions between

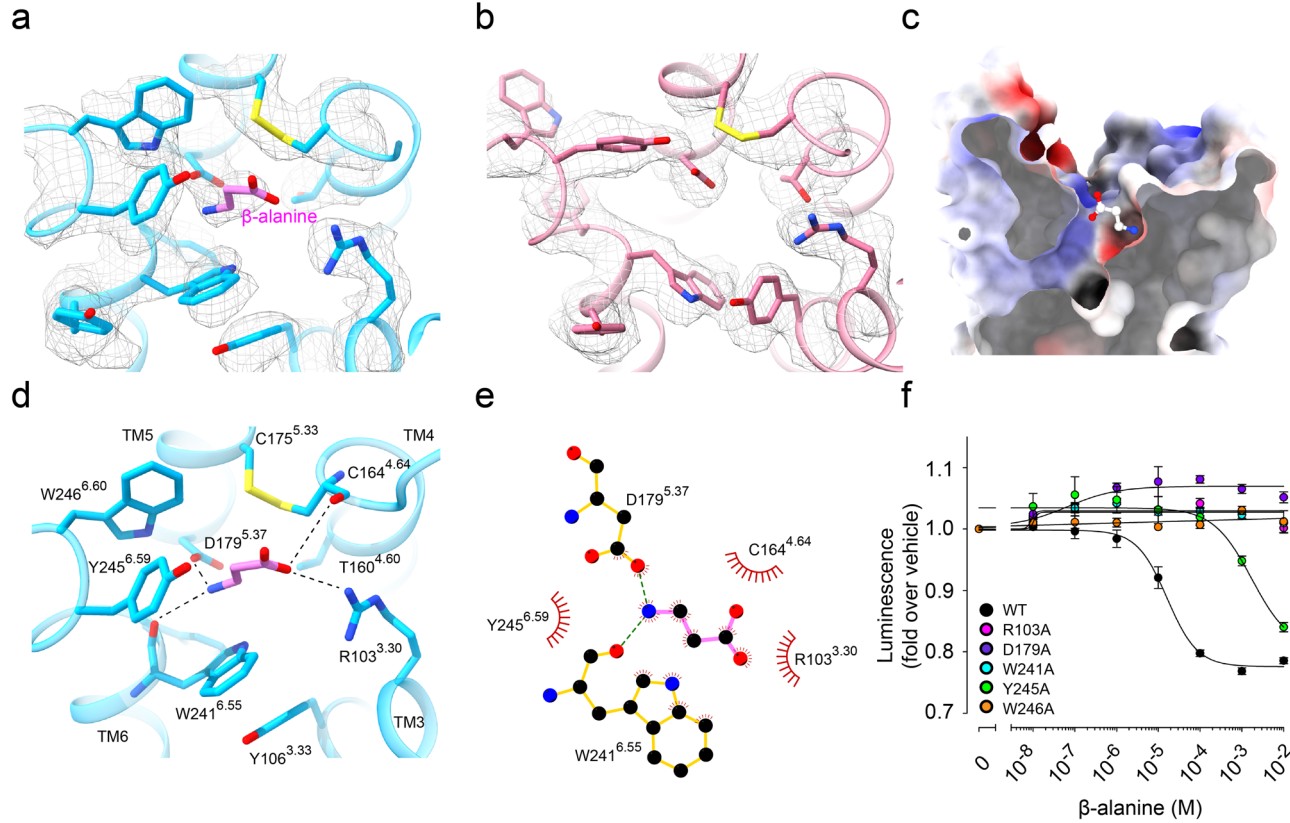

**Fig. 2 Ligand recognition of MrgD. a**, **b** Cryo-EM maps and models of the ligand-binding pocket of β-alanine (**a**) and apo MrgD (**b**) structures. The residues around the ligand-binding pocket and β-alanine are shown as stick models. The cryo-EM density maps are shown as a gray mesh. **c** Electrostatic surface potential of MrgD ligand-binding pocket in complex with β-alanine. β-alanine is shown as a stick model. **d** Ligand-binding pocket of β-alanine–MrgD. β-alanine and the side chains of MrgD contributing to ligand-binding are shown as stick models. Hydrogen bonds are shown as black dashed lines. **e** Schematic representation of β-alanine-binding interactions. Interactions were determined using LigPlot+. **f** MrgD-mediated Gi activation plot measured by the NanoBiT-G-protein dissociation assay. Dose–response curves are shown as means ± s.e.m. (standard error of the mean) of six independent experiments for WT and three for the mutants. Data for the graphs in **f** are available as Supplementary Data 1.

**Table 2 NanoBiT G protein dissociation assay.**

| Construct | β-alanine | |
|---|---|---|
| | pEC$_{50}$ ± s.e.m. | $E_{max}$ ± s.e.m. |
| WT | 4.85 ± 0.10(6) | 0.78 ± 0.007 |
| Cryo-EM construct | 5.16 ± 0.03(3) | 0.832 ± 0.01 |
| R103A | nd | nd |
| Y106A | 3.14 ± 0.03(3) | 0.874 ± 0.01 |
| Y106F | 3.38 ± 0.01(3) | 0.734 ± 0.01 |
| Y109A | 3.15 ± 0.02(3) | 0.698 ± 0.01 |
| Y109F | 3.99 ± 0.03(3) | 0.501 ± 0.01 |
| C164S | nd | nd |
| C175S | nd | nd |
| D179A | nd | nd |
| S234A | 5.11 ± 0.10(3) | 0.88 ± 0.007 |
| L237A | 3.03 ± 0.03 (3) | 0.847 ± 0.02 |
| Y240A | 4.16 ± 0.13(3) | 0.807 ± 0.01 |
| Y240F | 4.99 ± 0.10(3) | 0.819 ± 0.01 |
| W241A | nd | nd |
| W241F | 3.14 ± 0.08(3) | 0.883 ± 0.02 |
| Y245A | 2.77 ± 0.11(3) | 0.803 ± 0.1 |
| W246A | nd | nd |

NanoBiT-G-protein dissociation assay of mutant MrgD receptors.

ICL2 and Gi may be one of the factors that contribute to the large displacement of αN helix in Gi-bound GPCRs.

**Apo state and β-alanine-bound state of MrgD.** The overall structures in the apo and β-alanine-bound states are similar, but the ligand-binding pocket shows a marked difference (Fig. 6a). In the apo state, the side chain of W241$^{6.55}$ faces the cytoplasmic side and is sandwiched between Y106$^{3.33}$ and Y109$^{3.36}$. This interaction stabilizes the close contact between TM3 and TM6 even in the absence of the ligand (Fig. 6b). The binding of β-alanine changes the rotamer of the side chain of W241$^{6.55}$, allowing W241$^{6.55}$ and Y106$^{3.33}$ to form a π–π stack. Y245$^{6.59}$ and W246$^{6.60}$ also change their rotamer orientations to interact with β-alanine. TM6 is slightly un-twisted, resulting in the entry of F242$^{6.52}$ between TM3 and TM6, thereby enhancing the hydrophobic interactions among the aromatic amino acids between TM3 and TM6 (Fig. 6b). The ligand-dependent signaling from the apo state of MrgD is triggered by a series of rearrangements of TM6 upon β-alanine binding.

We sought to obtain clues to understand the mechanism of the basal activity by performing mutagenesis experiments. We first measured the basal activity of point mutants of residues within the ligand-binding site, which are involved in the interaction between TM3 and TM6. However, no reduction in the basal activity was observed in any of the mutants (Supplementary Fig. 14, Supplementary Data 2). We then focused on the two prominent hydrogen bonds between the side chain of Y109$^{3.36}$ and the backbone carbonyl of S234$^{6.48}$ and the other between the side chains of S234$^{6.48}$ and S268$^{7.45}$ in both the apo and β-alanine-bound states (Fig. 6c). Measurement of the basal activity of the mutants of these residues revealed that the Y109A, Y106F, and S234A mutations reduced the basal activity (Fig. 6d and

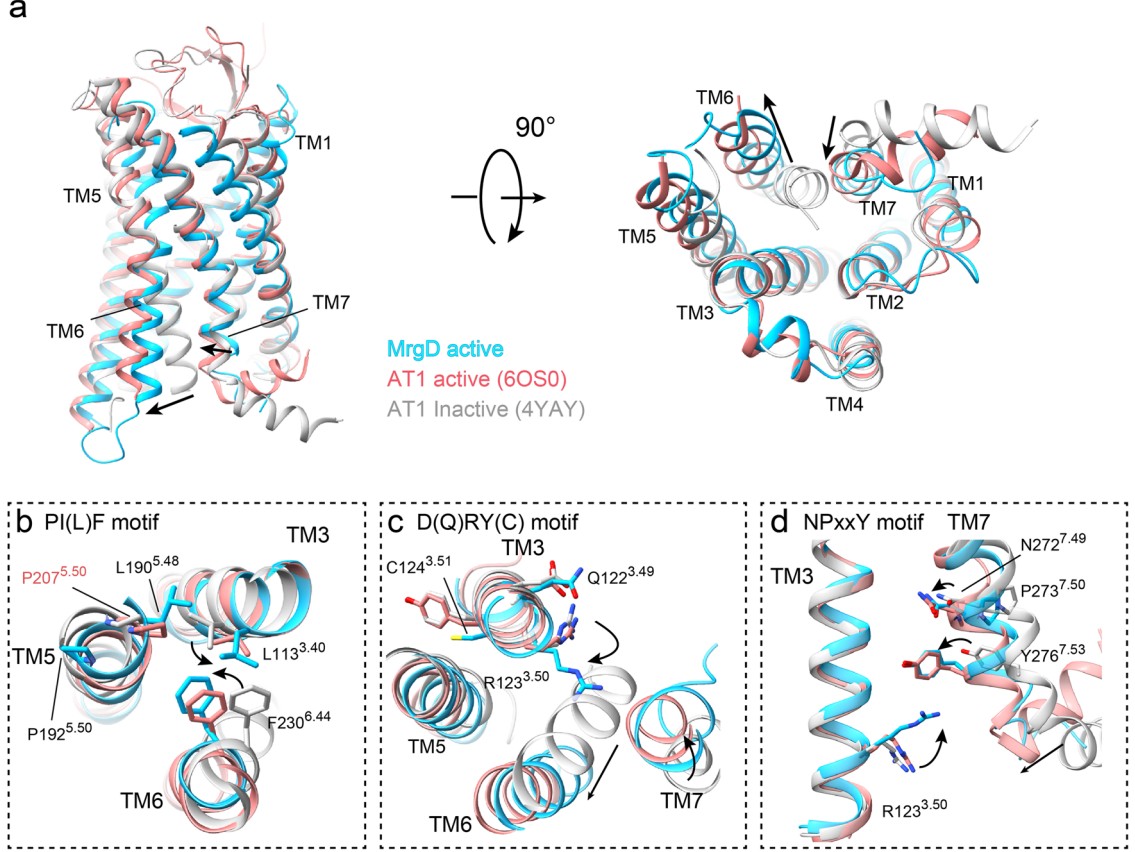

**Fig. 3 Active conformation of the MrgD receptor. a** Superposition of the β-alanine-bound active MrgD receptor (cyan), active Angiotensin II type1 (AT1) receptor (salmon, PDB code: 6OS0), and antagonist-bound AT1 receptor (gray, PDB code: 4YAY) models at two different angles. The movement of TM6 and TM7 in the MrgD receptor relative to inactive AT1R is indicated by black arrows. **b–d** Detailed views of the three motifs: PIF motif (**b**), DRY motif (**c**), and NPxxY motif (**d**). Each residue in the motifs is shown as a stick model. The difference in the orientation of the side chains from the inactive AT1R is shown in black arrows. The alignment is based on only the receptor portion.

Supplementary Data 2), suggesting that these two hydrogen bonds are involved in the basal activity of MrgD.

We also performed MD simulations to investigate the stability of the apo and β-alanine-bound states without G proteins. In the apo state, we observed closure of the cytoplasmic cavity of TM3-TM6 (Supplementary Fig. 15a, b). In the β-alanine-bound state, the distance between TM3 and TM6 on the cytoplasmic side remained constant for 1 μs and maintained the outward open conformation of TM6 (Supplementary Fig. 15a, b). In terms of the ligand-binding site, the conformation was more flexible, and the RMSD was much higher for the apo state than the β-alanine-bound state (Supplementary Fig. 15c). These results suggest that the binding of β-alanine and/or the G-proteins is necessary to maintain the outward open conformation.

## Discussion

This study demonstrates that the ligand-binding site of MrgD is a shallow pocket in the extracellular surface, and the bound ligand is exposed to the solvents. This binding mode is distinct from those observed in other class A GPCRs, in which the ligands are embedded in the deep binding pockets and isolated from the solvents.

Three notable structural features achieve this characteristic ligand-binding pocket of MrgD. First, the ECL2 of MrgD is short: only eight amino acids long. Typical ECL2 in class A GPCRs comprises 15–30 amino acids that cover the ligand-binding pocket and contribute to the ligand recognition and activation of GPCRs[47,48]. By contrast, the MrgD ECL2 is too short to cover

over the pocket, and the ligand bound to the pocket is exposed to the solvents (Supplementary Fig. 10). This feature is reminiscent of melanocortin receptors (MCR), which also have a short ECL2 and a large extracellular vestibule that allows for the binding of large peptide ligands[49].

The second feature is that MrgD, like CB1 and MCR, lacks $C^{3.25}$ of TM3 and $C^{45.50}$ of ELC2, although these are conserved in more than 90% of the class A GPCRs[50]. Instead, MrgD has a disulfide bond between $C164^{4.64}$ and $C175^{5.33}$, and the two cysteines are located near the binding pocket of β-alanine (Fig. 2a, d). Without being restraint by the disulfide bond forming TM3–ELC2, ECL2 of MrgD is flipped to the top of TM4 and TM5, increasing the solvent exposure tendency of the ligand-binding pocket. The results of the G protein dissociation assay in this study and the high conservation of these cysteine residues in the MRGPR family suggest that these cysteines may have been acquired during evolution as a necessary adaptation to form a ligand-binding pocket, allow proper folding, and enable receptor activation (Supplementary Fig. 11c).

The third feature is a 20-degree shift of TM6, which may hinder ligand binding to the classical class A orthosteric pocket. In addition, the difference in the kink in TM5 and the disulfide bonds may move the position of TM5 toward TM4, creating space for TM6 to tilt (Supplementary Fig. 10a). These features may provide ligand-binding pockets where ligands can efficiently bind and dissociate. The ligand-binding potency of β-alanine is very low, ranging from 4 to 20 μM, compared to other GPCRs agonists, which are high in potency (on the order of nM). The

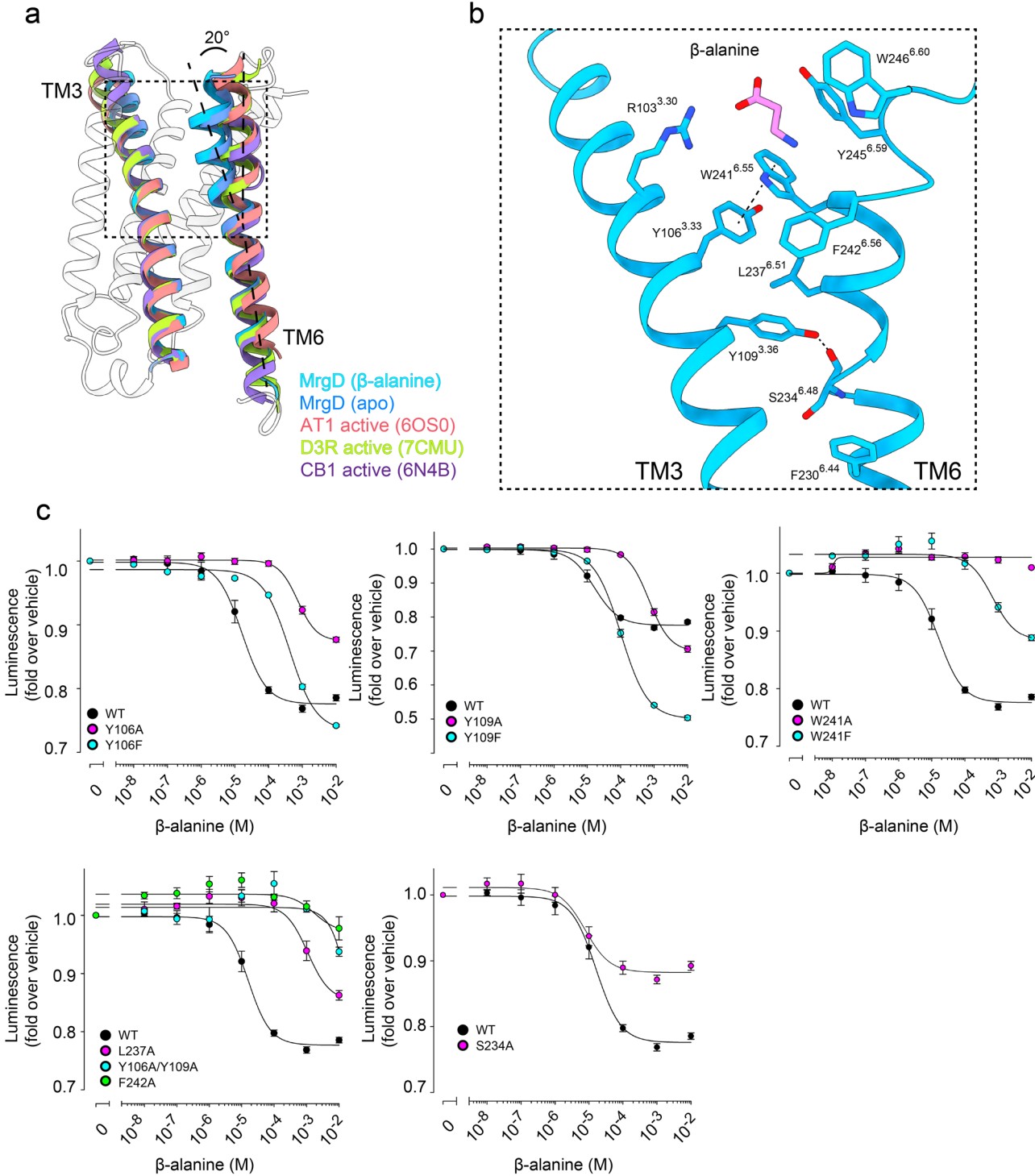

**Fig. 4 Unconventional activation of MrgD. a** Superposition of TM3 and TM6 of β-alanine-bound MrgD (cyan), apo-MrgD (blue), active AT1R[35] (salmon, PDB code: 6Os0), active D3R[37] (green, PDB code: 7CMU), and active CB1R[38] (purple, PDB code: 6N4B). For MrgD, the extracellular half of TM6 is tilted about 20° toward the TM3. **b** Magnified view of the area around the toggle switch on MrgD. The residues and ligands are shown as sticks. **c** NanoBiT–G-protein dissociation assay of the MrgD mutants for the residues in the extracellular half between TM3 and TM6. Dose-response curves are shown as means ± s.e.m. of six independent experiments for WT and three for the mutants. Data for the graphs in **c** are available as Supplementary Data 1.

concentration of β-alanine in the plasma of healthy humans is about 3.8 μM[51]. The shallow and solvent-exposed ligand-binding pocket may contribute to the low ligand-binding potency of β-alanine.

It has been reported that MrgD can also be activated by various ligands other than β-alanine. Previous studies have shown that GABA activates MrgD, albeit at a lower potency than β-

alanine[7,13], whereas L-alanine does not. These are all amino acids but differ in that the number of carbon atoms between the amino and carboxyl groups is one for L-alanine, two for β-alanine, and three for GABA (Supplementary Fig. 16). There are interactions between the amino group of β-alanine and D179 of MrgD and between the carboxyl group and R103 of MrgD; L-alanine would not activate MrgD because the formation of these

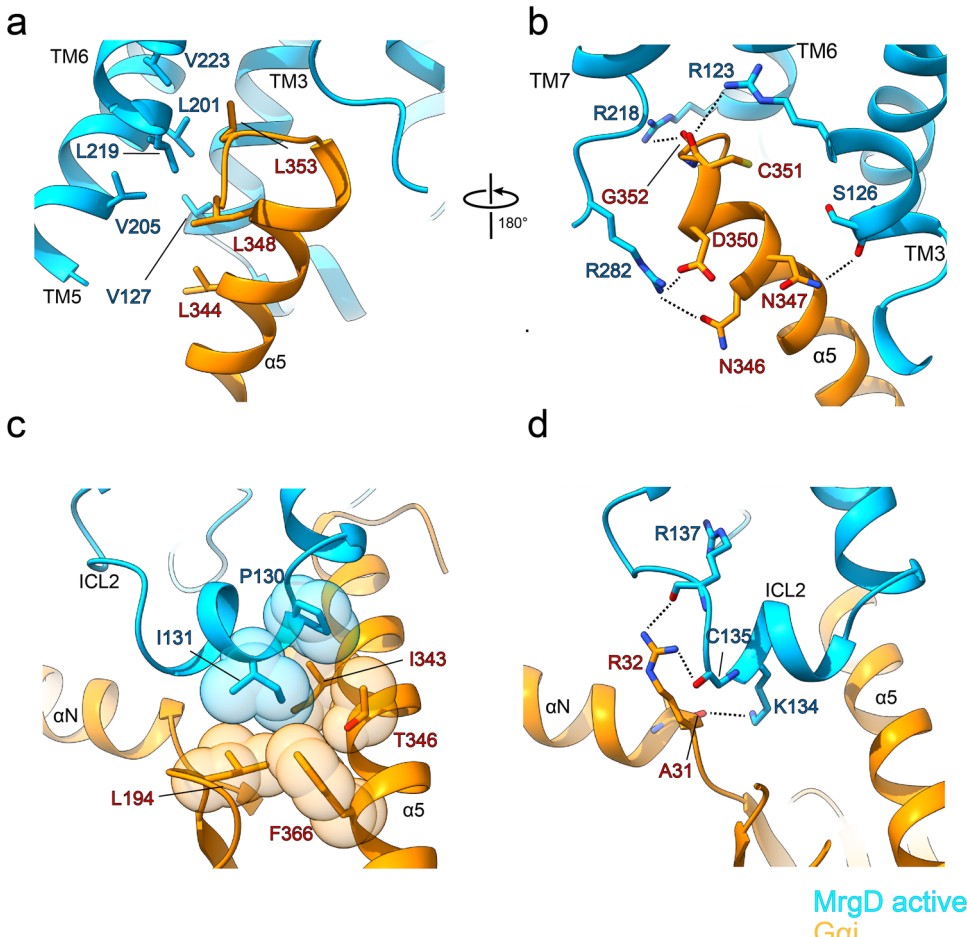

**Fig. 5 Binding interface between MrgD and G protein. a, b** Hydrophobic (**a**) and hydrophilic (**b**) interactions between MrgD and Gi around the α5 helix of Gi. **c, d** Hydrophobic (**c**) and hydrophilic (**d**) interactions between ICL2 of MrgD and the αN-α5 cleft of Gi. The residues involved in the interaction are shown as stick models. Black dashed lines indicate hydrogen bonds.

interactions is too short. GABA is long enough to access both amino acids but too long to make an optimal distance, resulting in low potency. Ang (1–7) and alamandine compete with β-alanine for binding to MrgD in vivo[52], suggesting that they share a similar binding position with β-alanine. Given the dynamic properties of the ligand-binding residues in MrgD (Fig. 6, Supplementary Fig. 16), MrgD may accommodate these high-affinity peptides by arranging orientations of those residues.

Many GPCRs show basal activity, but the details of the mechanism have not yet been established. In both the two states of MrgD, we see that $Y109^{3.33}$, $S234^{6.48}$, and $S268^{7.45}$ form a hydrogen-bond network (Fig. 6c). Our cAMP inhibition assay demonstrated that Y109A and S234A markedly reduced the basal activity of MrgD while still maintaining the activity induced by β-alanine (Fig. 6d). MrgD lacks the conserved sodium-binding site of TM3 (Supplementary Fig. 5), which is important for the stabilization of some GPCRs[36,53]. MrgD has a $Q^{3.49}$ rather than $D^{3.49}$ in the DRY motif, and $Q^{3.49}$ and $R^{3.50}$ may not be able to form the ion lock that is seen in the inactive state of other GPCRs[34,54]. In several GPCRs, mutations of $D^{3.49}$ show significantly increased basal activity[55,56]. These findings indicate that the absent sodium binding site and the modifications of the conserved motifs might destabilize the inactive state and shift the conformational equilibrium of the receptor from the inactive state to the active state populations.

The structures of MrgprX2 and MrgprX4 have recently been reported[57,58]. Good agreement is evident in the arrangement of

TM6 between these receptors and MrgD (Supplementary Fig. 17a). The binding position of the small agonist in MrgprX2 is consistent with that of MrgD (Supplementary Fig. 17b, c). However, the ligand selectivity differs: MrgprX2 uses the negatively charged amino acids $E^{4.60}$ and $D^{5.38}$ for ligand recognition, while MrgD uses one positively charged $R^{3.30}$ and one negatively charged $D^{5.37}$. The distribution of electrostatic potentials in the ligand-binding pocket also differs between the two (Supplementary Figs. 5 and 17d, e), raising a reasonable possibility that these receptors would recognize different ligands.

In the peptide-bound structure of MrgprX2, alamandine binds across two pockets in MrgprX2: one of these pockets corresponds to the β-alanine-binding site in MrgD. Assuming that MrgD is in complex with almandine, $D^{5.37}$ of MrgD would interact with R2 of alamandine, while Y4 and H6 of alamandine could interact with the empty pocket of MrgD (Supplementary Figs. 16, 17e). The ligand-binding pocket of MrgprX4 is positively charged (Supplementary Fig. 17f). MrgprX4 lacks a pocket corresponding to the β-alanine-binding site in MrgD, providing an explanation for the ligand-binding specificity of different MRGPR receptors.

In summary, our cryo-EM studies reveal the structural basis for the activation of MrgD induced by its endogenous agonist β-alanine. We also show a unique activation mechanism that lacks a toggle switch. No inactive state structures of MRGPR family proteins have not been reported yet. A high-resolution structural study of the inactivated state will be necessary to understand the

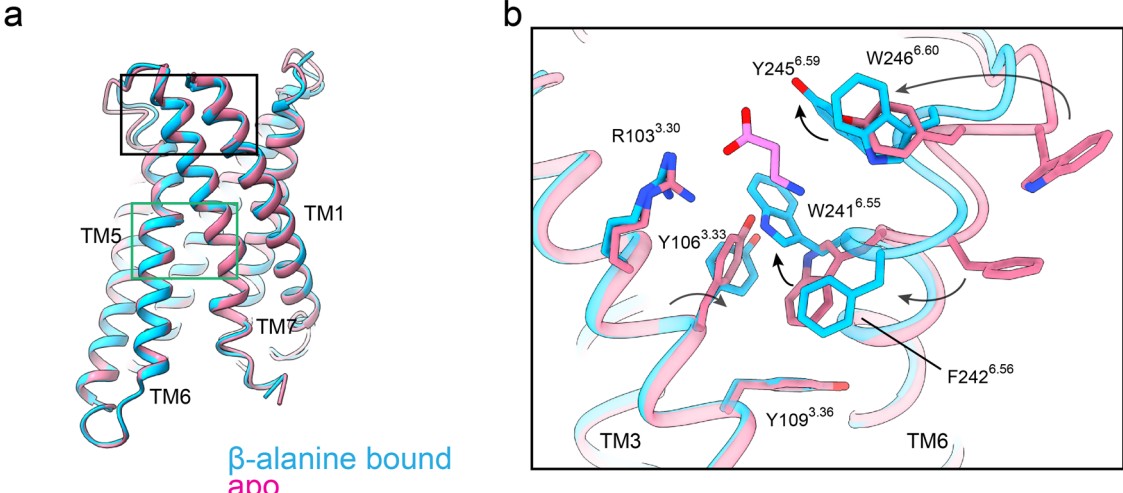

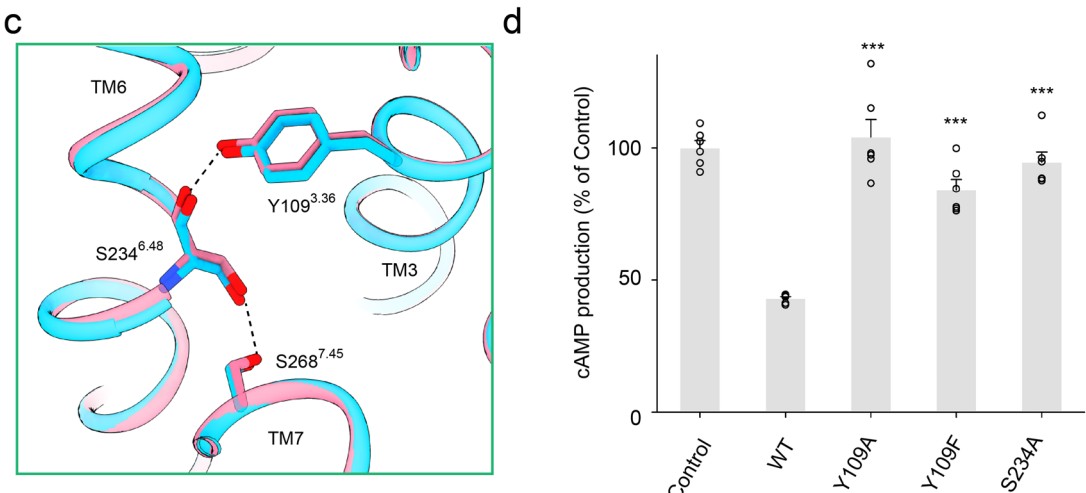

**Fig. 6 Apo vs. β-alanine-bound MrgD. a** Superposition of the apo state (pink) and β-alanine-bound state (cyan) of MrgD is represented in cartoon style. **b** Magnified view of the ligand-binding pocket of the apo (red) and β-alanine-bound MrgD (cyan). The residues of interest are shown as sticks. The black arrows indicate the conformational transition from the apo to the β-alanine-bound state. **c** Critical residues for the basal activity of MrgD The side chains of Y109$^{3.36}$, S234$^{6.48}$, and S268$^{7.45}$ are shown as sticks. The black dotted line indicates the hydrogen bond. Colors are consistent with (**b**). **d** The basal activity of WT MrgD and mutants was measured by cAMP inhibition assay. ***$p < 0.001$ (one-way analysis of variance [ANOVA] followed by the Dunnett's test, compared with the response of WT. Data represent mean ± s.e.m ($n = 6$). The expression levels of these mutants at the cell surface were comparable to the wild-type (Supplementary Fig. 6). Data for the graphs in **d** are available as Supplementary Data 2.

activation mechanism accurately. Our findings provide insights into the mechanism of MrgD signaling and will be helpful in drug discovery studies targeting the MRGPR family.

## Methods

**Constructs**. The N-terminal deleted human Mas-related G-protein-coupled receptor D (MrgD) (residues 5–321) (Uniprot: Q8TDS7) was sub-cloned into the pFastbac1 vector. Hemagglutinin signal peptide (HA) and FLAG-tag (DYKDDDDK), followed by the BRIL epitope, were fused at the N-terminus of MrgD to improve the expression level and receptor stability. Human Gαi1 (Gi) with two dominant-negative mutations (G203A, A326S), mouse wild-type Gβ1, Gγ2, and a single-chain antibody scFv16 (a kind gift from Dr. Brian Kobilka at Stanford University) were cloned into the pFastBac1 vector.

**Expression and purification of scFv16**. The His$_8$-tagged scFv16 was expressed and secreted by Sf9 insect cells[59]. The Sf9 cells were removed by centrifugation at 6200×$g$ for 10 min, and the supernatant was combined with 10 mM CaCl$_2$, 20 mM Tris (pH 8.0), 150 mM NaCl, and Protease Inhibitor Cocktail (Roche). The supernatant was mixed with Ni-NTA agarose resin (GE Healthcare Life Sciences) and stirred for 30 min at 4 °C. The collected resin was washed with buffer

containing 20 mM HEPES (pH 7.5), 500 mM NaCl, and 20 mM imidazole, and further washed with 10 column volumes of buffer containing 20 mM HEPES (pH 7.5), 100 mM NaCl, and 20 mM imidazole. The protein was eluted with 20 mM HEPES (pH 7.5), 100 mM NaCl, and 250 mM imidazole. The eluted fraction was dialyzed overnight at 4 °C. The purified scFv16 was concentrated to 5 mg mL$^{-1}$ and stored at −80 °C until use.

**Protein expression and purification**. MrgD, DNGi, Gβ1, Gγ2, and scFv16 were co-expressed in Sf9 insect cells using the Bac-to-Bac Baculovirus Expression System (Invitrogen). Cell cultures were grown in Sf-900™ II SFM medium (Gibco) to a density of 2–3 × 10$^6$ cell mL$^{-1}$ and then infected with the viruses expressing MrgD, DNGαi1, Gβ1, Gγ2, and scFv16. Cell culture was collected by centrifugation 48 h post-infection and stored at −80 °C.

Cell pellets were lysed by homogenization in 20 mM HEPES (pH 7.4), 50 mM NaCl and 10 mM MgCl$_2$, 10% glycerol, 25 mU mL$^{-1}$ apyrase (NEB), 1 mM β-alanine (Sigma-Aldrich), and Protease Inhibitor Cocktail (Roche). The apo state was purified without a ligand. After incubation at RT for 1 h, the membranes were solubilized by the addition of 0.5% (w/v) lauryl maltose neopentylglycol (LMNG, Anatrace) and 0.03% (w/v) cholesteryl hemisuccinate Tris salt (CHS, Anatrace) for 2 h at 4 °C. The supernatant was cleared by centrifugation and incubated with M2 FLAG resin (Sigma-Aldrich) for 1 h. After binding, the resin was washed with 10

column volumes of 20 mM HEPES (pH 7.4), 100 mM NaCl, 5 mM MgCl$_2$, 0.05% LMNG, 0.05% glyco-diosgenin (GDN, Anatrace), 0.003% (w/v) CHS, and 1 mM β-alanine. The complex was then washed with 10 column volumes of 20 mM HEPES (pH 7.4), 100 mM NaCl, 5 mM MgCl$_2$, 0.01% LMNG, 0.01% GDN, 0.0006% (w/v) CHS, and 1 mM β-alanine. The complex was then eluted in a wash buffer containing 200 μg mL$^{-1}$ FLAG peptide (Sigma-Aldrich). The eluted complex was supplemented with 100 μM TCEP (Fujifilm, Wako) for reducing conditions. The complex was purified by size-exclusion chromatography on a Superdex 200 10/300 column (GE) in 20 mM HEPES (pH 7.5), 100 mM NaCl, 1 mM β-alanine, 0.00075% LMNG, and 0.00025% GDN with 0.00004% CHS and 100 μM TCEP. Peak fractions were concentrated to ~15 mg mL$^{-1}$ for electron microscopy studies.

**Cryo-EM grid preparation and data collection**. The cryo-EM grids were prepared by applying 3.5 μL of the purified β-alanine-bound or apo MrgD–Gi complexes at ~15 mg mL$^{-1}$ to glow-discharged holey carbon grids (Quantifoil R1.2/1.3, Au, 300 mesh). The grids were plunge-frozen in liquid ethane using a Vitrobot Mark IV (Thermo Fisher Scientific). The frozen grids were transferred to liquid nitrogen and stored for data acquisition. Cryo-EM imaging was performed at the Institute for Protein Research, Osaka University, on a Titan Krios instrument (Thermo Fisher Scientific) operating at an acceleration voltage of 300 kV and equipped with a Cs corrector (CEOS, GmbH). Movies were recorded using a K3 detector (Gatan) in CDS mode at a magnification of 105,000× at the camera level, corresponding to a pixel size of 0.675 Å with 66 frames at a dose of 0.91 e$^-$/Å$^2$ per frame and an exposure time of 3 s per movie resulting in a total dose of 60 e$^-$/Å$^{-2}$ s$^{-1}$ with a defocus ranging from −0.7 to −1.9 μm. Data were automatically collected using the SerialEM software (https://bio3d.colorado.edu/SerialEM/) with an energy filter at a slit width of 20 eV. A total of 15,184 and 14,623 movies were collected for the β-alanine MrgD–Gi complex and the apo MrgD–Gi complex, respectively.

**Cryo-EM data processing**. Image stacks were subjected to beam-induced motion correction using MotionCor2.1[60]. The contrast transfer function (CTF) parameters for each non-dose-weighted micrograph were determined by CTFFIND-4[61]. Automated particle selection and data processing were performed using RELION-3.1[60]. For the dataset of the β-alanine MrgD–Gi complex, automated particle selection yielded 7.6 million particles. The particles were extracted on a binned dataset with a pixel size of 2.7 Å and were subjected to reference-free 2D classification, producing 2.1 million particles with well-defined averages. A 3D initial model was calculated with RELION, then low-pass-filtered to 20 Å and used as an initial reference model for 3D classification. The selected particles were re-extracted at a pixel size of 1.35 Å and subjected to 3D auto-refinement. We performed a masked 3D classification focusing on the TM domain. One class was subsequently subjected to 3D refinement, CTF refinement, and Bayesian polishing. The quality of the map was improved by repeating again the 3D classification without alignment. Finally, the 97,282 particles were subjected to 3D refinement, and the final map was determined at 3.1 Å.

For the dataset of the apo MrgD–Gi complex, automated particle selection yielded 8.7 million particles. The particles were extracted on a binned dataset with a pixel size of 2.7 Å and were subjected to reference-free 2D classification, producing 2.3 million particles with well-defined averages. The initial model was calculated from the two-dimensional average image using RELION. An initial reference model for 3D classification was prepared by low-pass-filtering to 20 Å, and this produced two good subsets showing clear structural features accounting for 1.2 million particles. Selected particles were re-extracted at a pixel size of 1.35 Å and subjected to 3D auto-refinement.

Since the obtained map was heterogeneous, we performed a focused classification masking TM. One distinct homogeneous map was obtained. Ultimately, 349,331 particles were subsequently subjected to 3D refinement, CTF refinement, and Bayesian polishing. The obtained particle set was processed by cryoSPARC[62] for non-uniform (NU) refinement[63]. The final map was determined at 2.8 Å. The reported resolution was based on the gold-standard Fourier shell correlation (FSC) using the 0.143 criterion[64]. Local resolution was determined in RELION 3.1 with half map reconstructions as inputs[60]. Further improvements to the two MrgD–Gi complex's map quality of the TM domain were obtained by performing local refinement in cryoSPARC.

**Model building and refinement**. The initial homology model of MrgD was a generated Swiss model using the AT1R as a template. The CB1R receptor coordinates (PDB 6N4B) were used as an initial model for the G-proteins, and the scFv16. Models were docked into the EM density map using UCSF Chimera[65]. This starting model was then subjected to iterative rounds of manual adjustment and automated refinement in Coot[66] and Phenix[67], respectively.

The statistics of the 3D reconstruction and model refinement are summarized in Table 1. All molecular graphics figures were prepared with UCSF ChimeraX[68].

**NanoBiT–G-protein dissociation assay**. The MrgD-induced Gi activation was measured with a NanoBiT–G-protein dissociation assay[31]. Specifically, a NanoBiT–Gi-protein consisting of a large fragment (LgBiT)-containing Gαi1 subunit and a small fragment (SmBiT)-fused Gγ2 subunit with the C68S mutation, along with the untagged Gβ1 subunit, was expressed with a test MrgD construct and the ligand-induced change in the luminescent signal was measured. The MrgD construct contained an N-terminal FLAG epitope tag (DYKDDDDK) of full-length human MrgD, and was inserted into the pcDNA3.1 vector. LgBiT was inserted between H.HA.29 and H.hahb.01 (common Gα numbering nomenclature[69]) of the Gi subunits flanked by 15-amino-acid flexible linkers (GGSGGGGSGGSSSGG) and cloned into the pcDNA3.1 vector. SmBiT was N-terminally fused to the Gγ2 subunit containing the C68S mutation with the 15-amino-acid linker and cloned into the pcDNA3.1.

HEK293 cells were seeded in a six-well culture plate at a cell density of $2 \times 10^5$ cells mL$^{-1}$ in 2 mL of complete DMEM and cultured in a humidified 37 °C incubator for 24 h with 5% CO$_2$. Plasmid transfection was performed in a six-well plate with a mixture of 100 ng Gi-Lg-encoding plasmid, 500 ng untagged-Gβ1-encoding plasmid, 500 ng smBiT Gγ2-encoding plasmid, and 200 ng GPCR-encoding plasmid (per well, hereafter). After 24 h of culture, the transfected cells were harvested with 1 mL of 0.53 mM EDTA-containing Dulbecco's PBS (D-PBS), followed by the addition of 2 mL of Hank's balanced salt solution (HBSS) containing HEPES. The cells were pelleted by centrifugation at 190×g for 5 min and resuspended in 2 mL HBSS containing 0.01% BSA and 5 mM HEPES (pH 7.4) (assay buffer). The cell suspension was seeded in a 96-well culture white plate (SPL Life Science) at a volume of 80 μL (per well hereafter) and loaded with 20 μL of 5× Nano-Glo Luciferase Assay Substrate (Promega) solution diluted in the assay buffer. After a 1 h incubation at RT, luminescent background signals were measured using a luminescence microplate reader (Multi-mode plate reader EnSpire, PerkinElmer). Test compound (6×, diluted in the assay buffer) was manually added to the cells (20 μL). Luminescent signals were measured 3–5 min after ligand addition and divided by the initial count. The ligand-induced signal ratio was normalized to that treated with the vehicle. The G-protein dissociation signals were fitted to a four-parameter sigmoidal concentration–response curve, from which the pEC$_{50}$ values (negative logarithmic values of half-maximum effective concentration (EC$_{50}$) values) and $E_{max}$ were used to calculate the mean and s.e.m.

**MD simulation**. Input models and parameters for all-atom MD simulations of the β-alanine-bound MrgD and the apo MrgD were prepared using CHARMM-GUI[70,71] and CHARMM-GUI Membrane Builder[72]. All the molecules except MrgD and β-alanine were removed from the cryo-EM models. Missing side chains were automatically modeled by CHARMM-GUI using the GalaxyFill algorithm[73]. Missing N-terminus and C-terminus of the MrgD were left as is. The protein residues were set to the standard CHARMM protonation states at neutral pH. The carboxy and amino groups of the β-alanine were de-protonated and protonated, respectively. The spatial arrangement of the MrgD in a lipid bilayer was determined using PPM server[74]. The MrgD was embedded in a lipid bilayer comprising 260 POPC molecules. The system was solvated and charge-neutralized by ~20,000 water molecules and ~100 mM NaCl. The system dimension and the total number of atoms were 100 Å × 100 Å × 109 Å, and 100,658 atoms for the β-alanine-bound MrgD, 100 Å × 100 Å × 106 Å, and 98,070 atoms for the apo MrgD. The CHARMM36m force-field parameters[75] were used for the proteins, lipids, and ions. The TIP3P model[76] was used for water. Topology and force-field parameter files for the β-alanine were generated using the CHARMM general force field (CGenFF)[77] and CHARMM-GUI Ligand Reader & Modeler[78].

MD simulations were performed using GROMACS 2020.3 and 2021.4[79]. First, the system was energy-minimized according to the protocol generated by CHARMM-GUI. Then, 3 independent MD runs were performed for both the β-alanine-bound MrgD and the apo MrgD. The run1 of the apo MrgD was performed using GROMACS 2020.3, and the others were performed using 2021.4. For each run, a 1 μs constant-NPT production run was performed after an equilibration run, according to the protocol generated by CHARMM-GUI. During the production run, the temperature and pressure were kept at 310.15 K and 1 bar using the Nosé-Hoover thermostat[80,81] and the Parrinello–Rahman barostat[82]. Bond lengths involving hydrogen atoms were constrained using the LINCS algorithm[83,84]. Long-range electrostatic interactions were calculated with the particle mesh Ewald method[85,86]. The simulations were carried out using the supercomputer "Flow" at Information Technology Center, Nagoya University. The trajectory analysis was performed using the MDAnalysis library[87,88].

For the simulations of the β-alanine-bound MrgD with the oppositely oriented β-alanine, the initial model was prepared by manually reversing the β-alanine orientation of the original β-alanine-bound MrgD model, followed by a sphere refinement with Coot to remove minor clashes. Three independent runs of MD simulations were performed with the same protocol as the original model, except that the final production runs were only performed for 250 ns.

**Cell-surface expression levels of MrgD receptor**. HEK293 cells were seeded in six-well plates at a cell density of $2 \times 10^5$ cells/well and cultured for 24 h. The cells were transfected with a complex of vector DNA (the FLAG-tagged MrgD receptor (control) or the indicated FLAG-tagged MrgD receptor mutants) and PEI. The next day, transfected HEK293 cells were harvested in PBS containing 0.53 mM EDTA, and then incubated in D-PBS containing 2% BSA and 2 mM EDTA (blocking buffer) for 30 min on ice. After centrifugation, the cells were stained with 10 μg mL$^{-1}$ anti-FLAG antibody (FUJIFILM Wako), followed 10 μg mL$^{-1}$ Alexa Fluor 488–conjugated anti-mouse IgG goat polyclonal antibody (Thermo Fisher Scientific) in blocking buffer.

The cells were washed once with D-PBS and resuspended in D-PBS containing 2 mM EDTA. Cell-surface expression levels were evaluated by flow cytometry on an Attune Flow Cytometer (Thermo Fisher Scientific). FLAG-positive cells were defined as cell populations with signals greater than the top 3% of MOCK cells.

**cAMP inhibition assay.** The inhibitory effects of different MrgD constructs or mutants in constitutive activity on forskolin-induced cAMP accumulation were measured using the GloSensor cAMP assay (Promega) according to the previous publications[89]. HEK293 cells were transiently co-transfected with the GloSensor and various mutants of MrgD or vehicle (pcDNA3.1) plasmids using PEI in six-well plates. After incubation at 37 °C for 24 h, the transfected cells were harvested with 0.53 mM EDTA-containing D-PBS, centrifuged at 190×g for 5 min, and suspended in HBSS containing 0.01% BSA and 5 mM HEPES (pH 7.4). Then, cells were resuspended in 2% GloSensor cAMP reagent at room temperature for 2 h. The cell suspension was seeded into a 96-well white plate (SPL life science) at a volume of 80 μL per well. A 20 μL volume of 50 μM forskolin diluted in HBSS containing HEPES and 0.01% BSA was added to each well (final, 10 μM), and the plates were incubated for 30 min at room temperature. The luminescence of the cells was measured using a microplate reader (Multi-mode plate reader EnSpire, PerkinElmer).

**Structure and sequence comparisons.** Sequence alignment was performed using the Clustal Omega server (https://www.ebi.ac.uk/Tools/msa/clustalo/) and the representation of sequence alignment was generated using the ESPript website (http://espript.ibcp.fr)[90]. The generic residue numbering of GPCR is based on the GPCRdb (https://gpcrdb.org/).

**Statistics and reproducibility.** All functional study data were analyzed using Sigma Plot (Systat) and presented as means ± s.e.m. from at least $n = 3$ experiments performed. Statistical differences were determined by one-way ANOVA with Dunnett's test.

**Reporting summary.** Further information on research design is available in the Nature Research Reporting Summary linked to this article.

## Data availability

Cryo-EM maps and atomic models for the β-alanine-bound-MrgD–Gi complex, apo-MrgD–Gi complex, β-alanine-bound-MrgD–Gi complex (local), and apo–MrgD complex (local) have been deposited in the Electron Microscopy Data Bank and wwPDB with the following accession numbers: EMD-33554 and PDB 7Y12, EMD-33557 and PDB 7Y15, EMD-33556 and PDB 7Y14, EMD-33555 and PDB 7Y13, respectively. The raw images have been deposited in the Electron Microscopy Public Image Archive, under accession code EMPIAR-11073 (beta-alanine bound MrgD) and EMPIAR-11074 (apo-MrgD). Source data underlying figures are presented in Supplementary Data 1 and 2. Uncropped versions of blots are presented in Supplementary Fig. 4.

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

## Acknowledgements

We are grateful to H. Iwao (JEOL) for maintaining the electron microscopes at Nagoya University. This work was supported by Grants-in-Aid for Scientific Research (B) under Grant Number 19H03165; Grant-in-Aid for Challenging Exploratory Research under Grant Number 21K19215; the Platform Project for Supporting Drug Discovery and Life Science Research (Basis for Supporting Innovative Drug Discovery and Life Science Research [BINDS]) from AMED (support number 2901) under Grant Numbers JP21am0101074; The Naito Foundation; Daiko Foundation; Foreign Young Invited Research Unit (B-2) (A.O.); JST PRESTO Grant Number JPMJPR21E5 (A.K.).

## Author contributions

S.S. and A.O. designed the research. S.S. and M.I. designed the expression constructs, purified the MrgD–Gi complex, and performed the NanoBiT assays of the MrgD mutants. S.S. performed cell surface expression in flow cytometry and cAMP inhibition assay. Y.H. helped with the preparation of scFv16. S.S., A.O., A.K., and T.K. collected the cryo-EM data using an automated data acquisition system. S.S. processed the cryo-EM data and built and refined the structure models. K.T. helped with data processing. K.T. performed the MD simulations. S.S. and A.O. wrote the manuscript. A.O. supervised the research. All authors read, revised, and approved the manuscript.

## Competing interests

The authors declare no competing interests.
