## [Peer Review File · Communications Biology]

Reviewers' comments:

Reviewer #1 (Remarks to the Author):

The manuscript "Structural insight into the activation mechanism of MrgD with heterotrimeric Gi-protein" by Shota Suzuki et al. targeted to determine the atomic-resolution cryo-EM structures of the apo MrgD with Gi complex and MrgD-Gi with its agonist β -alanine to reveal ligand-dependent activation of MrgD. Authors have calculated the high-resolution cryo-EM structures of apo and activated form of MrgD. Both the structures are 2.8Å and 3.1Å respectively, and both the structures are suitable for atomic model building. The authors also determined the ligand-binding pocket and performed a detailed structural analysis of apo MrgD and activated MrgD receptor.

This is a well-designed and attractive study with two high-resolution cryo-EM structures. There are no crystal or cryo-EM structures of apo-MrgD and MrgD with β -ala with G-protein complexes. Therefore, this is the first structure apo MrgD and activated MrgD receptor with G complex. However, I found some unclear findings; Data could have been represented better to support their conclusion and avoid many careless mistakes. My questions are listed below:

1. The "Introduction" is too short. Authors should pay attention and rewrite the introduction. Some incomplete sentences are there; e.g., the MRGPR family has been considered "orphans". " Is it orphans' receptors or orphans channels? Again Line 46, what is "Ang"? Line 47-48, "the physiological function In various cell signaling transduction."-- What kind of complications are there? It is not clear from the introduction. Line 43: "However, several receptors have been identified along with physiological ligands with" – proper original literature should be mentioned along with this review paper.
2. Authors mentioned Supplementary Fig. 2 on page number 6-line number 177. There is no Supplementary Fig. 2 in the manuscript. Additionally, authors have "Extended Data Fig.", I suppose these "Extended Data Files" are supplementary files. Then why do authors need separate Supplementary Figures? This is highly confusing to the reader. I think authors should use either Extended Data Files or supplemental files.
3. Figure 3 a-d numbers are not there. It is hard to follow the main text and correlate the figure numbers.
4. Could it be possible for the authors to highlight β -alanine density in EM Map in figure 1B or figure 2? In figure 2, the authors presented β -ala in mesh view. The authors could represent the entire cryo-EM map with β -ala in mesh view. At least from figure 1b and Figure 2a-b, it is not very clear the localization of β -ala with respect to the entire cryo-EM map. The author could add one extra figure of MrgD with β -ala in fig 2 and discuss the interactions of β -ala with other amino acids using Fig 2a-b. Additionally, in Extended Data Fig. 5, one amino acid (possible T) is completely out from the cryo-EM density map? The overall resolution of the map is very high; why does this amino acid outside of the density map? Is it possible that the above-mentioned amino acid has different stereochemistry? It will be better to incorporate an extra figure (supplemental) to show the bonding types between β -ala and water with TM.
5. Some noticeable amount of changes are observed in H8 in MrgD structure in the cytosolic region,

which is close to the inactive state of AT1 (4yay) in Figure 3a (probably fig 3a because figure numbers are not there). These changes are very significant. The authors should clarify this. Also, the PDB id should be uniform with text.

6. The authors have discussed a 20-degree shift of TM in figure 4a. However, the authors did not discuss this in the discussion. What is the biological significance of this 20-degree shift of TM?

7. Extended Data Fig. 6, based on the figure legends and figure, it is represented that authors compared MrgD with beta-ala, serotonin, ramelteon adenosine mdmb, but it is very clumsy and very difficult to understand.

8. Line 578: authors performed a masked 3D classification focusing on the TMs and part of the G protein. but in Extended Data Fig. 2 G protein part is not clear. Only the TM domain is visible. Again, in line 580: the authors performed 3D classification again after polishing without alignment. What is the reason to perform 3D classification again after polish? Again, if authors performed 3D classifications after particle polish, why they did not perform the rest of the steps? How many particles are there for 2nd time 3d classification? I think the authors performed CTF refine, auto-refinement again after the 2nd 3D classification. However, the authors did not mention this in the method section.

9. The authors compared their cryo-EM structure (apo & activated) with other published structures. It will be better to highlight it properly in the text and mention the PDB ID of these structures. One example is EP2-Gs.

Reviewer #2 (Remarks to the Author):

In this report by Suzuki and colleagues, the authors determine cryo-EM structures and analyze the activation of MrgD, a member of the Mas-related G protein-coupled receptors family of GPCRs involved in itch and nociception. In the presence of the weak agonist, b-alanine, MrgD adopts an activated state in which TM6 swings out, allowing Gi to bind. A similar conformation was resolved in the absence of ligand, suggesting that that MrgD does not need an agonist to access the activated state. Mrg family GPCRs lack the conserved tryptophan on TM6 that serves as a toggle during activation. In the b-alanine bound structure, the authors identify W241 as a possible alternative sensor that triggers a cascade via contacts between numerous hydrophobic residues that lead to activation, as was previously seen for the related MRGPX2 receptor in activated states. The critical roles of W241 and several of the residues involved in the hydrophobic cascade are validated through activation assays. Together, these analyses represent an important advancement in understanding mechanistic underpinnings of the receptor ligand recognition and activation. While the findings described in this report are well described, we have several suggestions for how this work can be improved.

Comments.

1. The ligand-binding pockets and nearby residues are one of the foci of the structural analyses presented in this report. However, the nearby extracellular loops are among the most poorly resolved regions of the protein. Several recent reports have taken advantage of focused refinements

to improve visualization of the extracellular regions of GPCRs in GPCR:G-protein complexes and it is likely that such an analysis would be beneficial for this study. Several of the modelled residues in the ECLs are barely visible in the maps provided to the reviewers. This is particularly important for MrgD because of its shallow orthosteric binding pocket, which places the ligands very near the extracellular surface of the receptor.

2. Continuing from the point 1, the authors model two densities in the unliganded structure as water molecules. These water molecules are proposed to bind in the orthosteric binding pocket near where b-alanine binds. At the moderate resolution of the unliganded map ($\sim 3.1\text{\AA}$), particularly in the flexibility of the extracellular region of the receptor, it is not clear from the map that these densities are indeed water molecules. As the authors suggest that these ordered water molecules may provide a mechanistic basis for the high basal activity of MrgD, additional evidence is required to demonstrate that these densities correspond to water molecules and not noise or a low-occupancy co-purified ligand.

3. The authors suggest that W241 recognizes the ligand in the orthosteric binding pocket and induces a tilt in TM6 through a concerted set of conformational switches. However, TM6 is in the activated conformation in apo structure without W241 undergoing a rotameric change. The authors need to more fully describe the role of W241 in the apo state and provide a rationale for how the receptor can be activated without changing the conformation of W241.

4. Structures of the related MRGPX2 in active states were recently reported. The authors briefly mention the existence of these reports, but it would be helpful to readers for the authors to compare their structures of MrgD with the MRGPX2 structures. The ligands used to determine the MRGPX2 structures are much larger and a comparison between the effect of small, weak agonists with more potent larger agonists would be informative for modeling how MrgD may be activated by larger agonists such as alamandine and Ang (1-7).

Minor Comments.

1. Inspection of the maps and models provided to the reviewers revealed several side chains that were incorrectly modeled. For example, Ga:R32 and Ga:Q333 are modelled as the incorrect rotamers in the b-alanine structure. These and other poorly modeled residues should be corrected prior to publication.

2. The residue W6.55 is mentioned in the last paragraph of the introduction without any prior description, making it hard for the general reader to understand its location or significance. The description of W6.55 should be left for the results section, where it can be properly introduced.

3. It is not clear which sample is presented in the gel filtration profile shown in ED Figure 1C. As there are few examples of unliganded GPCRs in complex with G-proteins, it would be helpful to show the profile in the apo state and the b-alanine bound state.

4. The scale for the local resolution plot in ED2 does not capture a useful dynamic range. A scale from 2.6 to 3.6 \AA would be a better representation of the buildable density, and would highlight the flexibility of the orthosteric ligand binding pocket. A per-residue CC plot from the output of real-space refinement (Phenix) would highlight the regions of lower model certainty, which is important for readers interpreting the results.

Reviewer #3 (Remarks to the Author):

In the manuscript "Structural insight into the activation mechanism of MrgD with heterotrimeric Gi-

protein” by Suzuki et al., the authors describe two cryo-EM structures of the Mas-related G protein-coupled receptor MrgD in complex with the heterotrimeric G protein Gi in the absence and presence of the ligand beta-alanine. MrgD has multiple physiological roles, including the regulation of blood pressure, neuropathic pain as well as perception of itching. Based on the structures, mutagenesis and NanoBiT-G protein activation assays, the authors postulate mechanisms for the ligand-dependent and constitutive activation of MrgD that shows variations or completely misses conserved microswitches present in other classA GPCRs. The structures presented in this manuscript together with the recently published structures of MRGPRX2 and MRGPRX4 provide important insights into the activation mechanism of these atypical GPCRs and would be of interest for the entire GPCR field. The manuscript is nicely illustrated and the results are overall clearly presented. However, a couple of major issues needs to be addressed before I can recommend this manuscript or publication in Communications Biology.

Major points:

Line 107: it seems like that the amide group of beta-alanine also forms a hydrogen bond with the backbone carbonyl of W241. This should be added to the description of the protein-ligand interactions.

Line 113 and Fig. 2c: What is the reason for the increase in luminescence in case of the mutant D179A ?

Line 124: It should be mentioned that the conserved disulfide bridge found in other GPCRs between TM3 and ECL2 is missing in the MrgD.

Line 133: Why is the MrgD compared to the AT1R receptor ? What is the RMSD between the structures and the sequence identity ?

Line 193: The authors show that mutation of S234 to Gly did not affect the G protein signaling activity. However, a glycine residue at this position could just provide enough flexibility in order to allow movement of the extracellular half of TM6 towards TM3. This might also explain the small left shift in the potency shown in the G protein activation assay. Did the authors also try to mutate the serin by the more apolar alanine ? This could destabilize the polar interaction between S234 and S268, which might be important for stabilizing the active state.

Ext. Fig. 10: Why does Y109F and also Y106F show higher Bmax values in the G protein activation assay compared to the WT ? They seem to stabilize the inward shifted extracellular part of TM6 in the ligand-bound conformation. Can the authors comment on this effect ?

Lines 224-232: This paragraph is very hard to understand. Please, check the language and rephrase the text to make your point more clear.

Line 224: R32 in the apo state-Gi complex does not seem to interact with the backbone atoms of R137 in contrast to the beta-alanine-bound MrgD-Gi complex. This suggests that this interaction might not be essential for G protein coupling. Can the authors comment on this ?

Paragraph 246-255: In order to see whether cholesterol modulates the signaling activity of MrgD, the binding site should be mutated to disrupt cholesterol-protein interactions using hydrophobic and bulky residues (Phe, Ile).

Line 309-311: The authors postulate that the structural waters in the ligand binding site in the absence of any ligands mimic the ligands and might be responsible for the high basal activity of the receptor. How does this agree with the mutational analysis of e.g. D179A which is involved in binding of one water molecule within the orthosteric ligand binding site, but does not show any effect on the basal activity of the receptor shown in fig. 2c. T160 should also be mutated to Ala to see if this water-binding site has an impact on the basal activity of the receptor.

Minor points:

Y245 in Fig. 2 should be labeled with the correct BW-number: 6.59 not 6.58

Line 153: Please, change Q131 to Q122, R132 to R123 and C133 to C124. This needs to be corrected in Fig. 3 as well

Line 235: Please, rephrase this sentence

Line 240-241: Please, rephrase this sentence.

Line 280: Please, rephrase this sentence.

Please, check the English language. There are many grammatical errors that makes the manuscript in parts hard to understand.

Reviewer #4 (Remarks to the Author):

The manuscript by Suzuki et al. presents cryoEM structures of the active MrgD GPCR in complex with Gi protein in apo- and b-alanine bound state. This study clearly illustrates binding mode of the endogenous ligand b-alanine to MrgD and the unique architecture of MrgD including alternative PIF motif, lack of canonical toggle switch, and shallow ligand binding pocket compared to other canonical class A GPCRs. The high-resolution information enabled to visualize the presence of the water molecules in the ligand binding pocket in the apo-state that might mimic the agonist to induce the high constitutive activity of MrgD. Comparison of the apo- and b-alanine bound states provides mechanistic insights into the activation through the unwinding of the upper part of the TM6. Mrgpr family is a relatively new GPCRs, and for many of them physiological function and endogenous ligand remain elusive. MrgD is one of the few to have been deorphanized to date and this report by Suzuki et al. will be a hallmark of the structural study to facilitate rational reagent design on this important class of GPCRs.

While this study provides compelling data on the activation mechanism and ligand binding mode of MrgD by presenting structural analysis and cell based signaling combined with mutations, I have several comments on the manuscript.

Page2 in the introduction, close to the bottom

“In addition, W6.55 recognizes a ligand directly and induces the tilt of TM6, leading to stabilization of the active form.”

Based on the Fig6b and the structures, it looks like Y6.59 and W6.60 are the main source of this TM transition rather than W6.55, while this residue make some contribution too.

Page3, in the first paragraph

“The harvested cells were incubated with β -alanine, an endogenous agonist, and apyrase to remove guanine nucleotides.”

Isn't it the membrane fraction, not cells, that were incubated with b-alanine?

Page3, the following paragraph

Please clarify apo-state complex was obtained from another preparation/dataset (from the supplementally fig and the methods, this looks like the case)

Page3, the paragraph of the ligand binding pocket of the MrgD receptor
With MrgprX structures bound with other ligands available, do the authors have any comments or discussion with regard to the ligand binding selectivity?

Page4, upper paragraph

I believe it is the backbone carbonyl of C164 of TM4, not C175 of TM5 that is within the range of hydrogen bonds with b-alanine. Please confirm this.

Page4, middle part

“... and the binding site is near ECL2, away from W6.48, which is characteristic of MrgD (Extended Data Fig.6).”

This W6.48 is not conserved in the MRGPR family. Could this be S6.48?

Page4,

The authors concluded that the disulfide bond makes essential contribution to the receptor activation. Given the apparent role of disulfide for the folding of the protein, it may be the protein stability/expression aspect that gave virtually no signal, not the receptor activation. Do authors have any idea how these mutations influence the protein expression or stability?

Page4, lower paragraph

The authors made a comparison of MrgD with AT1R to obtain structural insights into the activation mechanism. Is the AT1R the highest ranked in the structural homology? Is AT1R a plausible homologous model to MrgD? What is the rationale to use AT1R as a reference structure?

Page5,

“The kink may prevent ligand binding to a typical class A orthosteric site (Extended Data Fig.8).”

I do agree that the kink may contribute to the restraint of the ligand binding pocket but isn't it more likely that relatively closed TMs (especially TM6) and residues closer to the extracellular side contribute to this inability of the ligand to reach the canonical orthosteric site? The authors indeed discuss this constriction by TM3-TM6 at the beginning of the following section.

Page5, around the end of the upper paragraph

The authors compared the position of R3.50 between active MrgD and inactive AT1R, and conclude that the extended side chain of R3.50 is a result of the activation of MrgD, assuming that inactive AT1R represents inactive MrgD conformation. However, this could be a speculation unless structure of the MrgD inactive conformation is available or the authors have a plausible explanation to use AT1R as a reference model. Furthermore, AT1R in this nanobody-bound model doesn't show this extended R3.50 upon activation, yet the authors are using AT1R as a reference structure. Again, is there any particular reason using AT1R? or could there be another reference structure?

Page6, middle of the second paragraph

The authors claim that TM6 movement upon activation induces the downward orientation of S234. However, in the apo-state, S234 also points downward in the same way as b-Alanine state. How does the authors justify that this position of S234 is triggered by the b-alanine binding? Also, corresponding W in D3R is actually pointing upward, which is not consistent with the case of MrgD.

Page6,

“A position of 3.36 of TM3 is also known to regulate GPCR activation”

Would the authors be able to add a reference that indicates or summarizes this?

Page6,

“The rotameric state of Y1093.36 is consistent with the orientation of the inactive state of C1143.36 in D3R (Fig.4c).”

The figure shows “active” state.

Page6, last paragraph before the next section

In the MrgprX2 structure, the homologous Y113F mutation greatly reduced both the efficacy and EC50. Y109F in MrgD likely loses the hydrogen bond interaction with S234 backbone. While Y109F shows lower EC50 value than the WT, it gives much larger Emax than the WT. Do the authors have any idea of this behavior of the mutant such as the expression level, and the role of Y109 in the receptor activation?

Page7. Bottom

Regarding the untwisting model of the activation of MrgD, the cryoEM density largely supports the side-chain positions around the b-alanine bound state, but that of apo-state has some ambiguity, especially W246 density which is mostly missing. How can authors justify their untwisting TM6 model by Y245 and W246 under this situation?

Page8. Regarding the discussion about cholesterol.

Please check the geometry of the cholesterol molecules. Have a look of ones in other models and correct accordingly.

The cryoEM density of the second cholesterol the authors focus is relatively poor. In addition, the density of cholesterol at the interface between L190 is rather fragmented, indicating that the “interaction” between L190 and the cholesterol is not stable. How do the authors explain this under the context of the claim that this cholesterol is a possible PAM supporting the position of L190?

In the discussion section, the authors proposed that the water molecules bound in the apo-state are likely the cause of the high constitutive activity. Apart from this possibility, is there any structural feature the authors can describe that might contribute to the constitutive activity?

Hereafter, some comments about the figures.

Labels are missing in Fig.3 (a-d).

In Fig.3 caption, the authors mention that “the movements of side chains are shown as black arrows upon receptor activation”. Because there is no inactive state MrgD structure available, this “movement” actually represents comparison between the inactive AT1R and the active MrgD position, not conformational change within a single receptor. More appropriate phrasing should be placed.

Throughout the figures, the PDB code for D3R is assigned as 7CNU, but it should be 7CMU.

In Extended Data Fig.7 and the Table2, the authors labeled mutants as C164S and C175S. However, in the main text, it is described as C164A and C175A respectively. Please confirm and correct them.

Response to Reviews:

Specific answers to Reviewer #1 comments

Reviewer #1 (Remarks to the Author):

The manuscript “Structural insight into the activation mechanism of MrgD with heterotrimeric Gi-protein” by Shota Suzuki et al. targeted to determine the atomic-resolution cryo-EM structures of the apo MrgD with Gi complex and MrgD-Gi with its agonist β -alanine to reveal ligand-dependent activation of MrgD. Authors have calculated the high-resolution cryo-EM structures of apo and activated form of MrgD. Both the structures are 2.8Å and 3.1Å respectively, and both the structures are suitable for atomic model building. The authors also determined the ligand-binding pocket and performed a detailed structural analysis of apo MrgD and activated MrgD receptor.

This is a well-designed and attractive study with two high-resolution cryo-EM structures. There are no crystal or cryo-EM structures of apo-MrgD and MrgD with β -ala with G-protein complexes. Therefore, this is the first structure apo MrgD and activated MrgD receptor with G complex. However, I found some unclear findings; Data could have been represented better to support their conclusion and avoid many careless mistakes. My questions are listed below:

Authors

We thank this reviewer for the time, effort, and insightful comments on our manuscript, which certainly improve the quality of this manuscript.

1. The “Introduction” is too short. Authors should pay attention and rewrite the introduction. Some incomplete sentences are there; e.g., the MRGPR family has been considered “orphans”. Is it orphans’ receptors or orphans channels? Again Line 46, what is “Ang”? Line 47-48, “ the physiological function In various cell signaling transduction.” -- What kind of complications are there? It is not clear from the introduction. Line 43: “However, several receptors have been identified along with physiological ligands with” – proper original literature should be mentioned along with this review paper.

Authors:

We have revised the introduction carefully.

Line 42 “orphan receptors” instead of “orphan”.

Line 46 “Angiotensin (Ang)” instead of “Ang”.

Line 47-48 “the physiological function~In various cell signaling transduction.”

This sentence has been removed because it is ambiguous, and we have described it more concretely in the Introduction section.

Line 43: “However, several receptors have been identified along with physiological ligands with”

We have added the following citations to the references.

Steele, H. R. & Han, L. The signaling pathway and polymorphisms of Mrgprs. *Neurosci. Lett.* 744, 135562 (2021).

Karhu, T. et al. Isolation of new ligands for orphan receptor MRGPRX1 hemorphins LVV-H7 and VV-H7. *Peptides* 96, 61–66 (2017).

Shinohara, T. et al. Identification of a G Protein-coupled Receptor Specifically Responsive to β -Alanine*. 279, 23559–23564 (2004).

2. Authors mentioned Supplementary Fig. 2 on page number 6-line number 177. There is no Supplementary Fig. 2 in the manuscript. Additionally, authors have “;Supplementary Fig.”; I suppose these “;Extended Data Files”; are supplementary files. Then why do authors need separate Supplementary Figures? This is highly confusing to the reader. I think authors should use either Extended Data Files or supplemental files.

Authors:

We apologize for confusing readers. Those have been combined as Supplementary Fig.s.

3. Figure 3 a-d numbers are not there. It is hard to follow the main text and correlate the figure numbers.

Authors:

We apologize for the mistake. The numbering of Fig. 3 has been corrected.

4. Could it be possible for the authors to highlight β -alanine density in EM Map in figure 1B or figure 2? In figure 2, the authors presented β -ala in mesh view. The authors could represent the entire cryo-EM map with β -ala in mesh view. At least from figure 1b and Figure 2a-b, it is not very clear the localization of β -ala with respect to the entire cryo-EM map. The author could add one extra figure of MrgD with β -ala in fig 2 and discuss the interactions of β -ala with other amino acids using Fig 2a-b. Additionally, in Supplementary Fig. 5, one amino acid

(possible T) is completely out from the cryo-EM density map? The overall resolution of the map is very high; why does this amino acid outside of the density map? Is it possible that the above-mentioned amino acid has different stereochemistry? It will be better to incorporate an extra figure (supplemental) to show the bonding types between β -ala and water with TM.

Authors:

About the assignment of water molecules in the pocket of the apo state and the role of water molecules in the basal activity, we have decided to remove them from our model because we could not have obtained concrete evidence, and the manuscript has been revised accordingly.

We have updated Fig. 2 to include the superimposition of the EM map and the model (Fig. 2a.b).

Based on our model, we discussed the interactions of β -alanine with surrounding amino acids (Fig. 2d-f)

D179 had been entirely off the cryo-EM density map because this extracellular region has very high flexibility. We employed local refinement resulting in improved density and more precise modeling. Now, D179 fits nicely into the density map (Fig. 2b)

5. Some noticeable amount of changes are observed in H8 in MrgD structure in the cytosolic region, which is close to the inactive state of AT1 (4yay) in Figure 3a (probably fig 3a because figure numbers are not there). These changes are very significant. The authors should clarify this. Also, the PDB id should be uniform with text.

Authors:

As mentioned, approximately half of H8 is helix-like, which is oriented similar to the inactive AT1. However, our density map for H8 is poorly resolved, and the density corresponding to H8 is not visible in the recently reported structures of Mrgprx2 and MrgprX4 either (Cao et al. (2021), Yang et al. (2021)). H8 is not resolved due to its flexibility, or the MRGPR family proteins may not form a helical structure corresponding to H8 of AT1. Therefore, we would like to avoid discussing the conformation of H8 in this manuscript.

We have unified the PDB id in text and figures to uppercase according to the suggestion.

6. The authors have discussed a 20-degree shift of TM in figure 4a. However, the authors did not discuss this in the discussion. What is the biological significance of this 20-degree shift of TM?

Authors:

Honestly, the biological significance of a 20-degree shift of TM6 is unclear. We think that this tilt hinders ligand binding to the typical class A orthosteric binding site, creates the shallow binding pocket exposed solvent, and allows recognition of several ligands. We added the following sentence in the discussion section.

Page 10, line 321

The third feature is a 20-degree shift of TM6, which may hinder ligand binding to the classical class A orthosteric pocket. In addition, the difference in the kinks in TM5 and the disulfide bonds may move the position of TM5 toward TM4, creating space for TM6 to tilt (Supplementary Fig. 9a).

7. Supplementary Fig. 6, based on the figure legends and figure, it is represented that authors compared MrgD with beta-ala, serotonin, ramelteon adenosine mdmb, but it is very clumsy and very difficult to understand.

Authors:

We have revised the figure to separately show each structure. (The figure number has also been changed to Supplementary Fig. 5.)

Supplementary Fig. 8 | Ligand-binding positions of GPCRs

Agonists bound to MrgD and other class A GPCRs are shown as cartoon models. β -alanine (cyan, MrgD), ramelteon (pink, MT1R, PDB code: 7DB6), pramipexole (light green, D3R, PDB code: 7CMU), MDMB-Fubinaca (purple, CB1R, PDB code: 6N4B), adenosine (cadet blue, A1R, PDB code: 7LD4), serotonin (light orange, 5HT_{1A} PDB code: 7E2Y), and the residues at position 6.48 are shown as sphere models. (a) side view, (b) top view

8. Line 578: authors performed a masked 3D classification focusing on the TMs and part of the G protein. but in Supplementary Fig.. 2 G protein part is not clear. Only the TM domain is visible.

Authors:

Honestly, the TM domain and part of the G protein are visible distinctly in Supplementary Fig. 2. Please see Response Figure. 1 below.

Response Figure 1. 3D focus classification on the TM domain and part of G protein

The 2nd 3D classification of β -alanine binding states in Supplementary Fig. 2. is shown. The TM domain and the part of the G protein are indicated by black brackets, respectively.

Again, in line 580: the authors performed 3D classification again after polishing without alignment. What is the reason to perform 3D classification again after polish? Again, if authors performed 3D classifications after particle polish, why they did not perform the rest of the steps?

Authors:

3D classification without alignment after polishing is to sort out low-quality particles. The first polishing was enough to obtain a good quality map, and the second polish was omitted.

How many particles are there for 2nd time 3d classification?

Authors:

We used approximately 830,000 particles for the second 3D classification and approximately 25,000 particles for the third 3D classification (after polish). The finally selected particle number was 97,282 particles. These are mentioned in Supplementary Fig. 2.

I think the authors performed CTF refine, auto-refinement again after the 2nd 3D classification. However, the authors did not mention this in the method section.

Authors:

This study does not include CTF-refinement and polishing after no-alignment 3D-classification because those did not improve the map. We modified the Methods section as follows.

Page 19, line 694

The quality of the map was improved by repeating again the 3D classification without alignment. Finally, the 97,282 particles were subjected to 3D refinement, and the final map was determined at 3.1Å.

9. The authors compared their cryo-EM structure (apo & activated) with other published structures. It will be better to highlight it properly in the text and mention the PDB ID of these structures. One example is EP2-Gs.

Authors:

We revised the manuscript to indicate the PDB IDs of the compared structures when the structures appear in the text for the first time. Since the comparison with EP2 is not essential in this study, we have removed EP2 from the discussion.

Specific answers to Reviewer #2 comments

Reviewer #2 (Remarks to the Author):

In this report by Suzuki and colleagues, the authors determine cryo-EM structures and analyze the activation of MrgD, a member of the Mas-related G protein-coupled receptors family of GPCRs involved in itch and nociception. In the presence of the weak agonist, b-alanine, MrgD adopts an activated state in which TM6 swings out, allowing Gi to bind. A similar conformation was resolved in the absence of ligand, suggesting that that MrgD does not need an agonist to access the activated state. Mrg family GPCRs lack the conserved tryptophan on TM6 that serves as a toggle during activation. In the b-alanine bound structure, the authors identify W241 as a possible alternative sensor that triggers a cascade via contacts between numerous hydrophobic residues that lead to activation, as was previously seen for the related MRGPX2 receptor in activated states. The critical roles of W241 and several of the residues involved in the hydrophobic cascade are validated through activation assays.

Together, these analyses represent an important advancement in understanding mechanistic underpinnings of the receptor ligand recognition and activation. While the findings described in this report are well described, we have several suggestions for how this work can be improved.

Authors

We thank this reviewer for the time, effort, and insightful comments on our manuscript, which certainly improve the quality of this manuscript.

1. The ligand-binding pockets and nearby residues are one of the foci of the structural analyses presented in this report. However, the nearby extracellular loops are among the most poorly resolved regions of the protein. Several recent reports have taken advantage of focused refinement to improve visualization of the extracellular regions of GPCRs in GPCR:G-protein complexes and it is likely that such an analysis would be beneficial for this study. Several of the modelled residues in the ECLs are barely visible in the maps provided to the reviewers. This is particularly important for MrgD because of its shallow orthosteric binding pocket, which places the ligands very near the extracellular surface of the receptor.

Authors:

We performed additional local refinement using softmasks focusing on the TM domain. The resulting maps became higher quality and enabled us to build more reliable atomic models of β -alanine and ECLs (Response Figure. 2).

Response Figure 2. Local refined density map and model

Superimposition of the EM map and model after the local refinement. Around the ligand binding pocket, especially ECL2 and ECL3 are well resolved. Since some residues are still ill-resolved, we haven't modeled the sidechains for those residues. β -alanine bound state (left), apo state (right)

2. Continuing from the point 1, the authors model two densities in the unliganded structure as water molecules. These water molecules are proposed to bind in the orthosteric binding pocket near where β -alanine binds. At the moderate resolution of the unliganded map ($\sim 3.1\text{\AA}$), particularly in the flexibility of the extracellular region of the receptor, it is not clear from the map that these densities are indeed water molecules. As the authors suggest that these ordered water molecules may provide a mechanistic basis for the high basal activity of MrgD, additional evidence is required to demonstrate that these densities correspond to water molecules and not noise or a low-occupancy co-purified ligand.

Authors:

We performed MD simulations to confirm the water dynamics in the ligand-binding pocket. We also performed cAMP inhibition assays of the ligand-binding pocket mutants to see whether those residues affect the basal activity of MrgD. The MD simulations

showed that many water molecules enter the ligand-binding pocket of MrgD (Response Figure. 3), but no stable water-binding site existed. We also found that the alanine mutants of the ligand-binding pocket do not affect the basal activity of MrgD (Supplementary Fig. 11). These results suggest that water molecules in the ligand-binding pocket are not related to the basal activity of MrgD. Currently, there is no evidence for waters to trigger the basal activity, and the discussion regarding this point has been removed. Accordingly, we have removed waters from the atomic models.

Response Figure 3 Water molecules of the ligand-binding pocket observed in the molecular dynamics simulations.

a, Superimposed apo MrgD ligand-binding pocket and cryo-EM maps. The residues constituting the ligand-binding pocket are shown as stick models. EM map is shown in red mesh.

b, A snapshot of molecular dynamics simulations shows water molecules enter the ligand binding pocket. Water molecules are shown as stick models.

3. The authors suggest that W241 recognizes the ligand in the orthosteric binding pocket and induces a tilt in TM6 through a concerted set of conformational switches. However, TM6 is in the activated conformation in apo structure without W241 undergoing a rotameric change. The authors need to more fully describe the role of W241 in the apo state and provide a rationale for how the receptor can be activated without changing the conformation of W241.

Authors:

Thank you for pointing it out. Based on basal activity measurements, we admit that our original statements regarding W241 were incorrect.

We generated several alanine-substitution mutants of the ligand-binding pocket residues and measured the basal activities using a cAMP inhibition assay. We found that the residues around the ligand-binding pocket, including W241, did not affect the basal activity (Supplementary Fig. 11b). However, W241A was essential for β -alanine-dependent receptor activation (Fig. 2f). Therefore, the role of W241 is to recognize β -alanine, rearrange its rotamer, and form a π - π stacking with Y106 of TM3, stabilizing the TM6 tilt toward TM3 on the extracellular side (Fig. 6b).

We modified the following related sentences in the results section.

Page 7, line 229

These results support an essential function for the bulky hydrophobic residues in the extracellular half of TM3 and TM6 in the β -alanine dependent activation of MrgD.

Page 8, line 274

The ligand-dependent signaling from the apo state of MrgD is triggered by a series of rearrangements of TM6 upon β -alanine binding.

4. Structures of the related MRGPX2 in active states were recently reported. The authors briefly mention the existence of these reports, but it would be helpful to readers for the authors to compare their structures of MrgD with the MRGPX2 structures. The ligands used to determine the MRGPX2 structures are much larger and a comparison between the effect of small, weak agonists with more potent larger agonists would be informative for modeling how MrgD may be activated by larger agonists such as alamandine and Ang (1-7).

Authors:

We added the structure comparison and discussion with MrgprX in Supplementary Fig. 14.

The following paragraphs have been included in the discussion section to describe the size and charge distributions of ligands.

Page 11, line 357

The structures of MrgprX2 and MrgprX4 have recently been reported^{58,59}. Good agreement is evident in the arrangement of TM6 among these receptors and MrgD

(Supplementary Fig. 15a). The binding position of the small agonist in MrgprX2 is consistent with that of MrgD (Supplementary Fig 15b,c). However, the ligand selectivity differs: MrgprX2 uses the negatively charged amino acids E^{4.60} and D^{5.38} for ligand recognition, while MrgD uses one positively charged R^{3.30} and one negatively charged D^{5.37}. The distribution of electrostatic potentials in the ligand-binding pocket also differs between the two (Supplementary Figs. 5 and 15d,e), raising a reasonable possibility that these receptors would recognize different ligands.

In the peptide-bound structure of MrgprX2, alamandine binds across two pockets in MrgprX2: one of these pockets corresponds to the β -alanine binding site in MrgD. Assuming that MrgD is in complex with alamandine, D^{5.37} of MrgD would interact with R2 of alamandine, while Y4 and H6 of alamandine could interact with the empty pocket of MrgD (Supplementary Figs. 14, 15e). The ligand-binding pocket of MrgprX4 is positively charged (Supplementary Fig. 15f). MrgprX4 lacks a pocket corresponding to the β -alanine binding site in MrgD, indicating the diversity of ligands in the MRGPR family.

Supplementary Fig. 15 | Structural features of the MRGPR family

a, Superposition of MrgD and MrgprX2 (orange, PDB code; 7S8O) and MrgprX4 (pink, PDB code; 7S8P). **b,c** Ligand-binding pocket. The side chains contributing to ligand binding are shown as stick models (b) MrgD, (c) MrgprX2. **d-f**, Electrostatic surface representation of the (d) MrgD, (e) MrgprX2, (f) MrgprX4 extracellular pocket calculated using the APBS plugin in PyMOL, with β -alanine shown as a green stick and (R)-ZINC-3573 as a yellow stick, Cortistatin-14 as a cyan stick, and MS47134 as a pink stick.

Minor Comments.

1. Inspection of the maps and models provided to the reviewers revealed several side chains that were incorrectly modeled. For example, Ga:R32 and Ga:Q333 are modelled as the incorrect rotamers in the b-alanine structure. These and other poorly modeled residues should be corrected prior to publication.

Authors:

We honestly say that the R32 side chain is correctly modeled in the β -alanine-bound structure. We found two separating densities in the apo state, and thus R32 was modeled as the alternative conformation (Response Figure. 4). As for other residues, we checked them carefully and correctly modeled them.

2. The residue W6.55 is mentioned in the last paragraph of the introduction without any prior description, making it hard for the general reader to understand its location or significance. The description of W6.55 should be left for the results section, where it can be properly introduced.

Authors:

We avoided emphasizing W^{6.55}.

We changed this sentence as follows in the Introduction section.

Page 3, line 87

In addition, TM6 of MrgD significantly tilts toward the TM3 side upon ligand-binding, and a tight interaction of the bulky residues occurs between TM3 and TM6, leading to stabilizing the active conformation.

3. It is not clear which sample is presented in the gel filtration profile shown in ED Figure 1C. As there are few examples of unliganded GPCRs in complex with G-proteins, it would be helpful to show the profile in the apo state and the b-alanine bound state.

Authors:

The gel filtration profile shown in Fig. 1C (in the first version) was for the β -alanine bound state. We apologize for the confusion. The gel filtration profiles for the β -alanine-bound and apo states have been prepared separately. Please see Supplementary Fig. 2 and 3.

4. The scale for the local resolution plot in ED2 does not capture a useful dynamic range. A scale from 2.6 to 3.6 Å; would be a better representation of the buildable density, and would highlight the flexibility of the orthosteric ligand binding pocket. A per-residue CC plot from the output of real-space refinement (Phenix) would highlight the regions of lower model certainty, which is important for readers interpreting the results

Authors:

We agree that the original color scale was not so informative for readers. We changed a local resolution scale from 2.6 Å ~4.4 Å to capture the dynamic range (Supplementary Fig. 2e,3e). We used the same scale of local resolution as the high-resolution structures of MrgprX2 (Cao et al. (2022), Yang et al. (2022)).

We performed local refinement. The improved maps enabled the more reliable atomic modeling of β -alanine and ECLs (Response Figure. 2). The CC plots suggest that the models for the residues in ECL2 and ECL3 are sufficiently reliable. (Response Figure. 4). ECL1 is more flexible than ECL2 and ECL3, and the model is less reliable. Therefore, the discussion about ECL1 has been excluded from this paper.

Response Figure 2. Local refined density map and model

Superimposition of the EM map after local refinement and model. Around the ligand binding pocket, especially ECL2 and ECL3 are well resolved. Since some side chains are disordered, they are assigned alanine. β -alanine bound state (left), apo state (right)

CC plot (β -alanine bound state)

CC plot (Apo state)

Response Figure. 4 A per-residue CC plot from the output of real-space refinement CC plots for each residue extracted from the output of real-space refinement (Phenix) are shown. The three extracellular loops are highlighted in pink: β -alanine bound state (upper) apo state (bottom).

Specific answers to Reviewer #3 comments

Reviewer #3 (Remarks to the Author):

In the manuscript “;Structural insight into the activation mechanism of MrgD with heterotrimeric Gi-protein”; by Suzuki et al., the authors describe two cryo-EM structures of the Mas-related G protein-coupled receptor MrgD in complex with the heterotrimeric G protein Gi in the absence and presence of the ligand beta-alanine. MrgD has multiple physiological roles, including the regulation of blood pressure, neuropathic pain as well as perception of itching. Based on the structures, mutagenesis and NanoBiT-G protein activation assays, the authors postulate mechanisms for the ligand-dependent and constitutive activation of MrgD that shows variations or completely misses conserved microswitches present in other classA GPCRs. The structures presented in this manuscript together with the recently published structures of MRGPRX2 and MRGPRX4 provide important insights into the activation mechanism of these atypical GPCRs and would be of interest for the entire GPCR field.

The manuscript is nicely illustrated and the results are overall clearly presented. However, a couple of major issues needs to be addressed before I can recommend this manuscript or publication in Communications Biology.

Authors:

We thank this reviewer for the time, effort, and insightful comments on our manuscript, which certainly improve the quality of this manuscript.

Line 107: it seems like that the amide group of beta-alanine also forms a hydrogen bond with the backbone carbonyl of W241. This should be added to the description of the protein-ligand interactions.

Authors:

The β -alanine nitrogen atom forms a hydrogen bond with the W241 backbone. We updated the figures to depict the protein-ligand interactions (Fig. 2d and 2e).

Line 113 and Fig. 2c: What is the reason for the increase in luminescence in case of the mutant D179A ?

Authors:

D179A did not appear to affect the folding of the receptor, as its expression level at the cell surface was comparable to that of wild type (Supplementary Fig. 6). The C180A mutant of MrgX2 has also been reported to increase luminescence (Ext. Data Fig. 4 in Yang et al. (2022)). C180 is located near the ligand-binding pocket of MrgprX2. These mutations may cause an unexpected structural change in the ligand-binding pocket, but the detailed mechanism remains unknown.

Line 124: It should be mentioned that the conserved disulfide bridge found in other GPCRs between TM3 and ECL2 is missing in the MrgD.

Authors:

We added the descriptions in the discussion section as follows.

Page 9 line 311

The second feature is that MrgD, like CB1 and MCR, lacks C^{3.25} of TM3 and C^{45.50} of ELC2, although these are conserved in more than 90% of the class A GPCRs⁵². Instead, MrgD has a disulfide bond between C164^{4.64} and C175^{5.33}, and the two cysteines are located near the binding pocket of β -alanine (Fig. 2a,d).

Line 133: Why is the MrgD compared to the AT1R receptor? What is the RMSD between the structures and the sequence identity?

Authors:

We used AT1R because of the highest amino acid sequence homology among the existing class A GPCRs, except for the MRGPR family proteins. The RMSD with AT1R is 1.066 for the C α atoms, the sequence identity is 14%, and the sequence similarity is 34%. It is also suitable for comparison because MrgD has angiotensin II metabolites as physiological ligands.

Line 193: The authors show that mutation of S234 to Gly did not affect the G protein signaling activity. However, a glycine residue at this position could just provide enough flexibility in order to allow movement of the extracellular half of TM6 towards TM3. This might also explain the small left shift in the potency shown in the G protein activation assay. Did the authors also try to mutate the serin by the more apolar alanine? This could destabilize the polar interaction between S234 and S268, which might be important for stabilizing the active state.

Authors:

Based on the reviewer's comment, we now think that the result of S234G reflects unexpected main chain conformation change rather than an effect size of the serine residue. Because P236 is placed the one-turn helix upward of S234, this variant has too much influence on the flexibility or conformation of the helix. On the other hand, the potency was not affected when we performed a NanoBiT G protein dissociation assay about S234A, but Bmax was reduced (Fig.4c). The results indicate that the serine residue at this position could provide enough size to allow movement of the extracellular half of TM6 toward TM3 and/or that S234A prevents hydrogen bonds between S234 and S268, reducing the activity (Fig. 6c). For these reasons, S234A is enough to discuss the role of S234 in our manuscript, or rather, S234G may be thought to confuse the reader.

Therefore, we would like to take the S234A mutant and remove the S234G variant from the results section.

We have removed the sentences concerning S234G in the Results section and added the following sentences.

Page 7, line 215

To investigate whether the small side chain of S234^{6.48} can directly affect the rotamer of F230^{6.44}, we measured the Gi signaling activity of the S234A mutant. S234A did not affect potency but reduced efficacy (Fig. 4c, Supplementary Fig. 6, Table. 2), indicating that the serine residue at this position could provide enough size to allow movement of the extracellular half of TM6 toward TM3.

Page 9 line 279

We then focused on the two prominent hydrogen bonds between the side chain of Y109^{3.36} and the backbone carbonyl of S234^{6.48} and the other between the side chains of S234^{6.48} and S268^{7.45} in both the apo and β -alanine-bound states (Fig. 6c). Measurement of the basal activity of the mutants of these residues revealed that mutations, Y109A, Y106F, and S234A reduced the basal activity (Fig. 6d), suggesting that these two hydrogen bonds are essential for the basal activity of MrgD.

Ext. Fig. 10: Why does Y109F and also Y106F show higher Bmax values in the G protein activation assay compared to the WT? They seem to stabilize the inward shifted extracellular part of TM6 in the ligand-bound conformation. Can the authors comment on this effect?

Authors:

Although the mechanism for the increased activity is unclear, we think there are two possibilities.

First, the Substitution of tyrosine with phenylalanine may have increased the hydrophobicity and strengthened the hydrophobic interaction of the side chains between TM3 and TM6, resulting in a higher B_{max} (Fig. 4b, c).

Second, breaking the hydrogen bond with the backbone carbonyl may have increased the dynamics of TM6, making the TM3-TM6 cavity more accessible to G proteins.

Lines 224-232: This paragraph is very hard to understand. Please, check the language and rephrase the text to make your point more clear.

Authors:

We modified the sentence as follows.

Page 8 line 251

Neurotensin receptor 1 (NTSR1) (PDB code: 6OS9) and CB1 interact with Gi via only hydrophobic residues at position 34.51 (Supplementary Fig. 11b,c), whereas D3R and μ -opioid receptor (μ OR) (PDB code: 6DDE) interact with Gi via the polar residues of the ICL2. (Supplementary Fig. 11d,e). In melatonin receptor MT1 (PDB code: 7DB6)⁴⁷, ICL2 is far from the hydrophobic cleft of Gi (Supplementary Fig. 11f), while in CCR6 (PDB code: 6WWZ)⁴⁸, it is tightly attached to the hydrophobic cleft due to interactions created by F^{34.54} (Supplementary Fig. 11g). When the receptor portion of each complex is superimposed, the orientation of the α N helix of Gi is diverse (Supplementary Fig. 11h). Unlike Gs-bound GPCRs, the interface between the receptor and Gi is variable. The different interactions between ICL2 and Gi may be one of the factors that contribute to the large displacement of α N helix in Gi-bound GPCRs.

Line 224: R32 in the apo state-Gi complex does not seem to interact with the backbone atoms of R137 in contrast to the beta-alanine-bound MrgD-Gi complex. This suggests that this interaction might not be essential for G protein coupling. Can the authors comment on this ?

Authors:

We carefully checked our apo-state map and model and noticed that an alternative rotamer had been missed. One rotamer interacts with the backbone carbonyl of R137 in the same manner as the β -alanine-bound state (Response Figure. 5a,b), and the other has

no interaction with R137 (Response Figure. 5a). Therefore, it is possible that the interaction of R32 is not essential or has only a minor contribution to Gi-coupling, especially in the apo state.

Response Figure. 5 Alternative rotamers of R32 in apo MrgD-Gi complex

Interaction between MrgD and Gi subunit. EM density is shown as grey. R32 and R137(ICL2) are shown as a stick model. Apo MrgD is colored pink (left), and β -alanine-bound MrgD is colored cyan (right). Gi is colored orange. The dashed lines show hydrogen bonds.

Paragraph 246-255: In order to see whether Cholesterol modulates the signaling activity of MrgD, the binding site should be mutated to disrupt cholesterol-protein interactions using hydrophobic and bulky residues (Phe, Ile).

Authors:

We employed the local refinement to improve this cholesterol-like density. The obtained EM map suggests that this density may not be Cholesterol but palmitic acid (part of phosphatidylcholine). Accordingly, we have updated and replaced Cholesterol with Palmitic acid in our model. Currently, there is no direct evidence that palmitic acid plays a role in PAM. We have removed the text regarding the discussion on Cholesterol from the Results section.

Line 309-311: The authors postulate that the structural waters in the ligand binding site in the absence of any ligands mimic the ligands and might be responsible for the high basal activity of the receptor. How does this agree with the mutational analysis of e.g. D179A which is involved in binding of one water molecule within the orthosteric ligand binding site, but does not show any effect on the basal activity of the receptor shown in Fig. 2c. T160 should also be mutated to Ala to see if this water-binding site has an impact on the basal activity of the receptor.

Authors:

We acknowledge that we should have rigorously addressed the roles of the water molecules and the binding pocket residues in the basal activity. We performed MD simulations of the apo state, and basal-activity measurements of various mutants of the binding pocket residues. The results of the MD simulation showed that many water molecules enter the ligand-binding pocket of MrgD (Response Figure. 3), but no stable water-binding site existed. We also found that the alanine mutants of the ligand-binding pocket do not affect the basal activity of MrgD (Supplementary Fig. 11). These results suggest that water molecules are present in the ligand-binding pocket but do not affect the basal activity of the MrgD. Therefore, we concluded that there is no conclusive evidence for the waters triggering the basal activity, and the discussions regarding this point should be removed. We removed waters from the atomic models. The results and discussions about water molecules in the ligand-binding pocket have been removed from the manuscript.

Response Figure 3 Water molecules of the ligand-binding pocket observed in the molecular dynamics simulations.

- a**, Superimposed apo MrgD ligand-binding pocket and cryo-EM maps. The residues constituting the ligand-binding pocket is shown as stick models. EM map is shown in red mesh.
- b**, A snapshot of molecular dynamics simulations shows water molecules enter the ligand binding pocket. Water molecules are shown as stick models.

Minor points:

Y245 in Fig. 2 should be labeled with the correct BW-number: 6.59 not 6.58

Authors:

This has been fixed.

Line 153: Please, change Q131 to Q122, R132 to R123 and C133 to C124. This needs to be corrected in Fig. 3 as well

Authors:

We apologize for the mistake. The residue numbers have been corrected.

Line 235: Please, rephrase this sentence

Authors:

We have modified the text as follows.

Page 8, line 266

In the apo state, the side chain of W241^{6.55} faces the cytoplasmic side and is sandwiched between Y106^{3.33} and Y109^{3.36}. This interaction stabilizes the close contact between TM3 and TM6 even in the absence of the ligand (Fig. 6b).

Line 240-241: Please, rephrase this sentence.

Authors:

We have modified the text as follows.

Page 8, line 267

The binding of β -alanine changes the rotamer of the side chain of W241^{6.55}, allowing W241^{6.55} and Y106^{3.33} to form a π - π stack. Y245^{6.59} and W246^{6.60} also change their rotamer orientations to interact with β -alanine. TM6 is slightly un-twisted, resulting in the entry of F242^{6.52} in between TM3 and TM6, thereby enhancing the hydrophobic interactions among the aromatic amino acids between TM3 and TM6 (Fig. 6b).

Line 280: Please, rephrase this sentence.

Authors:

We have modified the text as follows.

Page 10, line 328

The unique ligand-binding pocket may be necessary for MrgD to recognize a stimulus of β -alanine above physiological concentrations.

Please, check the English language. There are many grammatical errors that makes the manuscript in parts hard to understand.

Authors:

The text was sent to an English proofreading service, and native English speakers checked the grammar.

Specific answers to Reviewer #4 comments

Reviewer #4 (Remarks to the Author)

The manuscript by Suzuki et al. presents cryoEM structures of the active MrgD GPCR in complex with Gi protein in apo- and β -alanine bound state. This study clearly illustrates binding mode of the endogenous ligand β -alanine to MrgD and the unique architecture of MrgD including alternative PIF motif, lack of canonical toggle switch, and shallow ligand binding pocket compared to other canonical class A GPCRs. The high-resolution information enabled to visualize the presence of the water molecules in the ligand binding pocket in the apo-state that might mimic the agonist to induce the high constitutive activity of MrgD. Comparison of the apo- and β -alanine bound states provides mechanistic insights into the activation through the unwinding of the upper part of the TM6. Mrgpr family is a relatively new GPCRs, and for many of them physiological function and endogenous ligand remain elusive. MrgD is one of the few to have been orphanized to date and this report by Suzuki et al. will be a hallmark of the structural study to facilitate rational reagent design on this important class of GPCRs.

While this study provides compelling data on the activation mechanism and ligand binding mode of MrgD by presenting structural analysis and cell based signaling combined with mutations, I have several comments on the manuscript.

Authors:

We thank this reviewer for the time, effort, and insightful comments on our manuscript, which certainly improve the quality of this manuscript.

Page2 in the introduction, close to the bottom

“;In addition, W6.55 recognizes a ligand directly and induces the tilt of TM6, leading to stabilization of the active form.”;

Based on the Fig6b and the structures, it looks like Y6.59 and W6.60 are the main source of this TM transition rather than W6.55, while this residue make some contribution too.

Authors:

We agree that Y^{6.59} and W^{6.60} contribute to the β -alanine dependent TM transition along with W^{6.55}. This is also consistent with our experimental results that W^{6.55}A and W^{6.60}A abolished the Gi activity (Fig. 4c), and Y^{6.59}A reduced Gi activity (Fig. 2f). The result suggests that W^{6.55}, Y^{6.59}, and W^{6.60} are essential for the proper β -alanine dependent Gi

signaling activity, although we could not have determined which residue has a stronger contribution. The text of the results section has been revised not to emphasize W^{6.55} as follows.

Page 4, line 141

These analyses suggest that R103^{3.30} and D179^{5.37} were required for binding to β -alanine, whereas W241^{6.55}, Y245^{6.59}, and W246^{6.60} play crucial roles in forming the ligand-binding pocket and activation of the receptor.

Page3, in the first paragraph

“;The harvested cells were incubated with β -alanine, an endogenous agonist, and apyrase to remove guanine nucleotides.”;

Isn't it the membrane fraction, not cells, that were incubated with b-alanine?

Authors:

We apologize for the confusion and have corrected “the membrane fraction”, not “cells”.

Page3, the following paragraph

Please clarify apo-state complex was obtained from another preparation/dataset (from the supplementally Fig and the methods, this looks like the case)

Authors:

The apo state complex was obtained from another preparation and dataset. We modified Supplementary Fig.2 and 3 to clearly show that the dataset and preparation of the apo state and β -alanine bound state are independent.

Page3, the paragraph of the ligand binding pocket of the MrgD receptor

With MrgprX structures bound with other ligands available, do the authors have any comments or discussion with regard to the ligand binding selectivity?

Authors:

We have updated the manuscript to discuss the differences between MrgD and MrgprXs. The paragraph below has been incorporated into the discussion section to describe the size and charge distributions of ligands. Please also check Supplementary Fig. 14.

Page 11, line 352

The structures of MrgprX2 and MrgprX4 have recently been reported^{58,59}. Good agreement is evident in the arrangement of TM6 among these receptors and MrgD (Supplementary Fig. 15a). The binding position of the small agonist in MrgprX2 is consistent with that of MrgD (Supplementary Fig. 15b,c). However, the ligand selectivity differs: MrgprX2 uses the negatively charged amino acids E^{4.60} and D^{5.38} for ligand recognition, while MrgD uses one positively charged R^{3.30} and one negatively charged D^{5.37}. The distribution of electrostatic potentials in the ligand-binding pocket also differs between the two (Supplementary Figs. 5 and 15d,e), raising a reasonable possibility that these receptors would recognize different ligands.

In the peptide-bound structure of MrgprX2, almandine binds across two pockets in MrgprX2: one of these pockets corresponds to the β -alanine binding site in MrgD. Assuming that MrgD is in complex with almandine, D^{5.37} of MrgD would interact with R2 of almandine, while Y4 and H6 of almandine could interact with the empty pocket of MrgD (Supplementary Figs. 14, 15e). The ligand-binding pocket of MrgprX4 is positively charged (Supplementary Fig. 15f). MrgprX4 lacks a pocket corresponding to the β -alanine binding site in MrgD, indicating the diversity of ligands in the MRGPR family.

Supplementary Fig. 15 | Structural features of the MRGPR family

a, Superposition of MrgD and MrgprX2 (orange, PDB code; 7S8O) and MrgprX4 (pink, PDB code; 7S8P). **b,c** Ligand-binding pocket. The side chains contributing to ligand binding are shown as stick models (b) MrgD, (c) MrgprX2. **d-f**, Electrostatic surface representation of the (d) MrgD, (e) MrgprX2, (f) MrgprX4 extracellular pocket calculated using the APBS plugin in PyMOL, with β -alanine shown as a green stick and (R)-ZINC-3573 as a yellow stick, Cortistatin-14 as a cyan stick, and MS47134 as a pink stick.

Page4, upper paragraph

I believe it is the backbone carbonyl of C164 of TM4, not C175 of TM5 that is within the range of hydrogen bonds with b-alanine. Please confirm this.

Authors:

We apologize for the mistake and have carefully corrected these in the revised manuscript.

Page4, middle part

“; and the binding site is near ECL2, away from W6.48, which is characteristic of MrgD (Supplementary Fig..6).”;

This W6.48 is not conserved in the MRGPR family. Could this be S6.48?

Authors:

We have corrected the mistake in the revised manuscript.

Page 5, line 147

The binding site is near ECL2, away from S^{6.48}, which is characteristic of MrgD (Supplementary Fig. 6).

Page4,

The authors concluded that the disulfide bond makes essential contribution to the receptor activation. Given the apparent role of disulfide for the folding of the protein, it may be the protein stability/expression aspect that gave virtually no signal, not the receptor activation. Do authors have any idea how these mutations influence the protein expression or stability?

Authors:

Thank you for pointing it out. The expression of the two mutants (C164S, C175S) on the cell surface was significantly reduced (Supplementary Fig. 6), suggesting that the disulfide bond is essential for the proper folding and trafficking of MrgD.

We have modified the sentence in the results section as follows.

Page 5, line 152

The expression level of these two mutants (C164S, C175S) on the cell surface was significantly reduced (Supplementary Fig. 5). These results indicate an essential contribution of the disulfide bond to proper folding and trafficking.

Page4, lower paragraph

The authors made a comparison of MrgD with AT1R to obtain structural insights into the activation mechanism. Is the AT1R the highest-ranked in the structural homology? Is AT1R a plausible homologous model to MrgD? What is the rationale to use AT1R as a reference structure?

Authors:

AT1R has the highest amino acid sequence homology (except for the MRGPR family); the sequence identity is 14%, and the sequence similarity is 34%. In addition, we think that AT1R is a plausible homologous model of MrgD because C α RMSD is as small as 1.066. It is also suitable for comparison because MrgD has angiotensin metabolites as physiological ligands.

Page5,

“;The kink may prevent ligand binding to a typical class A orthosteric site (Supplementary Fig..8).”;

I do agree that the kink may contribute to the restraint of the ligand binding pocket but isn't it more likely that relatively closed TMs (especially TM6) and residues closer to the extracellular side contribute to this inability of the ligand to reach the canonical orthosteric site? The authors indeed discuss this constriction by TM3-TM6 at the beginning of the following section.

Authors:

TM3-6 closure would probably be the direct cause of the blockage to reach the canonical orthosteric site (Supplementary Fig. 9).

We also think that the unique TM5-kink and disulfide bond in TM4-5 pull the position of TM5 toward TM4, creating a space for TM6 to tilt in (Supplementary Fig..8a).

For clarity, we have added the following sentences in the discussion section.

Page 10, line 321

The third feature is a 20-degree shift of TM6, which may hinder ligand binding to the classical class A orthosteric pocket. In addition, the difference in that kinks of TM5 and the disulfide bonds may move the position of TM5 toward TM4, creating space for TM6 to tilt (Supplementary Fig. 9a).

Page5, around the end of the upper paragraph

The authors compared the position of R3.50 between active MrgD and inactive AT1R, and conclude that the extended side chain of R3.50 is a result of the activation of MrgD, assuming that inactive AT1R represents inactive MrgD conformation. However, this could be a speculation unless structure of the MrgD inactive conformation is available or the authors have a plausible explanation to use AT1R as a reference model. Furthermore, AT1R in this nanobody-bound model doesn't show this extended R3.50 upon activation, yet the authors are using AT1R as a reference structure. Again, is there any particular reason using AT1R? or could there be another reference structure?

Authors:

AT1R shows the highest sequence similarity with MrgD in the class A GPCRs (the sequence identity is 14%, and the sequence similarity is 34%). We also checked structural alignments between β -alanine bound MrgD and various class A GPCRs in the active state: AT1R (PDB:6OSO), AT2R (PDB:6JOD), D3R (PDB:7CMU), and CB1 (6N4B). AT1R has the lowest C α RMSD against MrgD (1.066 for AT1R, 1.069 for AT2R, 1.01 for D3R, 1.103 for CB1). Furthermore, MrgD recognizes an endogenous ligand peptide alamandine, a metabolite of angiotensin II., and AT1R has been solved in active and inactive states, facilitating our discussion.

As pointed out, the active AT1R model does not show the extended R3.50. However, the active AT1R structure was solved without G proteins. Some active state GPCR structures have shown an intact R^{3.50}-D^{3.49} salt bridge as seen in the Nanobody bound AT1R (Rasmussen et al. (2011)). As far as we know, R3.50 is extended in all GPCR-G protein complexes reported so far. The absence of G-proteins in the active AT1R structure may cause the non-extended R3.50. Although the AT1R-G protein complex is not yet understood, we assume that AT1R, like other GPCRs, has an extended R3.50 conformation when in complex with G proteins.

Page6, middle of the second paragraph

The authors claim that TM6 movement upon activation induces the downward orientation of S234. However, in the apo-state, S234 also points downward in the same way as b-Alanine state. How does the authors justify that this position of S234 is triggered by the b-alanine binding?

Authors:

The orientation of the S234 and Y109 are downward in both states. We performed an additional NanoBiT- G protein dissociation assay and showed that S234A reduced the ligand-dependent activity. We also found that S234A, Y109A, and Y109F significantly reduced basal activity (Fig. 6d). These results suggest that the arrangement of S234 is downward without β -alanine binding.

We apologize that the related sentence was very confusing to the reviewer. We have modified the sentence as follows.

Page 6 line 208

The binding of β -alanine triggers the formation of π - π stacking between W241^{6.55} and Y106^{3.33}, stabilizing the TM3 interaction of TM6. Position 3.36 of TM3 is also known to regulate GPCR activation³⁹. The Y109^{3.36} faces TM6 and forms a hydrogen bond with the backbone carbonyl of S234^{6.48}, which stabilizes the active form. The side chains of Y109^{3.36} and S234^{6.48} in MrgD tilt toward the cytoplasmic side (Supplementary Fig. 10a). This is in contrast to the other class A GPCRs in which they are oriented to the extracellular side (Supplementary Fig. 10b-d).

Also, corresponding W in D3R is actually pointing upward, which is not consistent with the case of MrgD.

Authors:

Thank you for pointing it out. We admit that it was an inconsistent statement. We have revised the results section as follows.

Page 7, line 212

The side chains of Y109^{3.36} and S234^{6.48} in MrgD tilt toward the cytoplasmic side (Supplementary Fig. 10a). This is in contrast to the other class A GPCRs in which they are oriented to the extracellular side (Supplementary Fig. 10b-d).

Page6,

“;A position of 3.36 of TM3 is also known to regulate GPCR activation”;

Would the authors be able to add a reference that indicates or summarizes this?

Authors:

We have cited the following references in the text.

Venkatakrishnan, A. J. *et al.* Molecular signatures of G-protein-coupled receptors. *Nature* **494**, 185–194 (2013).

Page6,

“;The rotameric state of Y1093.36 is consistent with the orientation of the inactive state of C1143.36 in D3R (Fig.4c).”;

The figure shows “;active”; state.

Authors:

We changed the expression to the difference between MrgD and other typical class A GPCRs, rather than active or inactive. Fig. 4 has been updated, and we modified the sentence as follows.

Page 7, line 212

The side chains of Y109^{3.36} and S234^{6.48} in MrgD tilt toward the cytoplasmic side (Supplementary Fig. 10a). This is in contrast to the other class A GPCRs in which they are oriented to the extracellular side (Supplementary Fig. 10b-d).

Page6, last paragraph before the next section

In the MrgprX2 structure, the homologous Y113F mutation greatly reduced both the efficacy and EC50. Y109F in MrgD likely loses the hydrogen bond interaction with S234 backbone. While Y109F shows lower EC50 value than the WT, it gives much larger Emax than the WT. Do the authors have any idea of this behavior of the mutant such as the expression level, and the role of Y109 in the receptor activation?

Authors:

We do not have concrete evidence, but we think that the more hydrophobic property of Phe enhanced the interaction between TM3 and TM6, leading to a more potent activity. Unlike MrgprX2, the hydrophobic interactions there may be meaningful in MrgD activation. We have confirmed that the expression level of Y109F on the cell surface was comparable to WT (Supplementary Fig. 6).

Page7. Bottom

Regarding the untwisting model of the activation of MrgD, the cryoEM density largely supports the side-chain positions around the β -alanine bound state, but that of apo-state has some ambiguity, especially W246 density which is mostly missing. How can authors justify their untwisting TM6 model by Y245 and W246 under this situation?

Authors:

We acknowledge the relatively lower quality of those parts in our original map and model in the apo state. We performed additional local refinement using a softmask focusing on the TM domain. The improved maps allowed us to build a near atomic model for the ECLs, including W246 (Response Figure. 6). Additional NanoBIT-G-protein dissociation assay showed that F242A and W246A lost Gi activity (Fig. 2f). These results indicate that the conformational changes of F242, Y245, and W246 upon β -alanine binding are essential for the ligand-dependent activation of MrgD (Fig. 6b), consistent with our untwisting model.

Response Figure 6. Local refined density map and model focused extracellular side of TM6

Superimposition of the EM map after local refinement and model. Extracellular side of TM6 amino acids are well fit to the EM map, β -alanine-bound state (a), apo state (b). EM maps are shown as grey mesh and these model are shown as ribbon and stick.

Please check the geometry of the cholesterol molecules. Have a look of ones in other models and correct accordingly.

The cryoEM density of the second Cholesterol the authors focus is relatively poor. In addition, the density of Cholesterol at the interface between L190 is rather fragmented, indicating that the “interaction”; between L190 and the Cholesterol is not stable. How do the authors explain this under the context of the claim that this Cholesterol is a possible PAM supporting the position of L190?

Authors:

We employed the local refinement to improve this cholesterol-like density. The obtained EM map suggests that this density may not be Cholesterol but palmitic acid (part of phosphatidylcholine). Accordingly, we have updated and replaced Cholesterol with Palmitic acid in our model. Currently, there is no direct evidence that palmitic acid plays a role in PAM. We have removed the text regarding the discussion on Cholesterol from the Results section

In the discussion section, the authors proposed that the water molecules bound in the apo-state are likely the cause of the high constitutive activity. Apart from this possibility, is there any structural feature the authors can describe that might contribute to the constitutive activity?

Authors:

Based on our additional mutagenesis experiments and MD simulations, we now think that neither the water molecules nor the residues in the ligand-binding pocket contribute to the basal activity. We measured the basal activities of the mutants of the ligand-binding pocket and found no reduction in the basal activity (Supplementary Fig.11). MD simulations showed the high dynamics property of the waters in the ligand-binding pocket, suggesting no stable water-binding site exists. We have removed waters from the atomic models. In the revised manuscript, we have removed the descriptions and discussions about water molecules.

Instead, we found that Y106^{3.33} and S234^{6.48} are essential for basal activity (Fig. 6d). We conclude that these specific amino acids, but not water molecules, contribute to the basal activity. We added the following sentence to the discussion.

Page 11, line 345

Many GPCRs show basal activity, but the details of the mechanism have not yet been established. In both of the two states of MrgD, we see that Y109^{3,33}, S234^{6,48}, and S268^{7,45} form a hydrogen-bond network (Fig. 6c). Our cAMP inhibition assay demonstrated that Y109A and S234A markedly reduced the basal activity of MrgD while still maintaining the activity induced by β -alanine (Fig. 6d). MrgD lacks the conserved sodium binding site of TM3 (Supplementary Fig. 5), which is essential for stabilizing the inactive state^{36,55}. MrgD has Q^{3,49}, rather than D^{3,49} in the DRY motif, and Q^{3,49} and R^{3,50} may not be able to form the ion lock that is seen in the inactive states. In several GPCRs, mutants of D^{3,49} show significantly increased basal activity^{56,57}. These findings indicate that the inactive conformation of MrgD is not stable and can therefore assume the active conformation more quickly, resulting in higher basal activity.

Hereafter, some comments about the figures.

Labels are missing in Fig.3 (a-d).

Authors:

This has been fixed.

In Fig.3 caption, the authors mention that “;the movements of side chains are shown as black arrows upon receptor activation”;. Because there is no inactive state MrgD structure available, this “;movement”; actually represents comparison between the inactive AT1R and the active MrgD position, not conformational change within a single receptor. More appropriate phrasing should be placed.

Authors:

The legend has been revised as follows.

Page 27 line 917

The difference in the orientation of the side chains from the inactive AT1R is shown in black arrows.

Authors:

This has been fixed.

In Supplementary Fig..7 and the Table2, the authors labeled mutants as C164S and C175S. However, in the main text, it is described as C164A and C175A respectively. Please confirm and correct them.

Authors:

This has been fixed.

Reviewers' comments:

Reviewer #1 (Remarks to the Author):

The authors have modified the introduction portion significantly. The author performed more cryo-EM data processing, which improves the cryo-EM structures. Authors modified several figures and represented the figures more appropriately, which will be easier for the general reader. Additionally, the authors explained most of the reviewer's suggestions. The authors addressed most of the questions, and their responses are convincing. Therefore, in my opinion, this manuscript would be accepted for publication.

Reviewer #2 (Remarks to the Author):

In this revised manuscript by Sazuki and colleagues, the authors describe structures of MrgD in b-alanine and ligand-free states. The structures presented with the revised manuscript are of higher quality and provide a structural framework for investigating MrgD and related MRGPR family receptors. In response to reviewer comments, the authors substantially revised their interpretations of the structures. Despite the changes, several of the claims made by the authors are only weakly supported and additional changes to the text will be necessary prior to publication.

Comments:

1. In the abstract, the authors claim that their structures provide a structural basis for the high basal activity. While the authors do determine structures in ligand-bound and ligand-free activated states, the mechanism of ligand-independent activation is still not clear. The authors show that mutation of Y109 and S234 reduces ligand-independent activation. However, they show in figure 4 that these mutations also diminish ligand-dependent activation and thus it is unclear if these mutations simply break activation or are specific to basal activity.
2. The ligand-bound MD simulations were performed starting from the pose assigned based on the local chemical environment of the ligand as the density was insufficient for unambiguous modeling. Did the authors also attempt any alternative poses to determine if their approach could "rigorously evaluate the binding orientation of b-alanine"? Especially as the affinity of MrgD for b-alanine is quite low.
3. The MD analysis indicates that the activated conformation is less stable for the apo state. However, MrgD has basal activity. How do the dynamics of TM6 compare to receptors with low basal activity?

Minor points:

1. Introduction section between Line 46-60 is a little repetitive. They mention b-alanine activating Gi, Gq, and Gs activation at multiple nonsequential points. This section could be refined.
2. The introduction has been modified to shift focus even further to the apo conformation and its role in basal activity, but as discussed above, it is not clear what mechanistic insight can be drawn from their findings.
3. Between Line 102-107 they describe how they co-expressed and co-purified the complex using b-alanine but provide no information about how they generated the apo complex – it should be mentioned here as well (as discussed in M&M Line 644).

Reviewer #3 (Remarks to the Author):

The authors sufficiently addressed my previous concerns. The manuscript has been significantly improved by the addition of new mutational data, MD simulations and updated figures. However, I still have some comments regarding the role of S234 and the reason for the basal activity of the receptor. In the following I list some major and minor points the author should address before I can recommend publication of their manuscript:

Major:

lines 215-219: I am still not sure what the authors want to say here. The S234A mutation proves that the H-bond between this side chain and residue S268 is important for the stabilization of the active state, which has been now nicely addressed by the authors on Page9 line 279. However, in order to investigate the impact of the side chain size at this position on the reorientation of F230, the authors should also substitute the side chain with larger ones (e.g. Tyr or the canonical Trp). This is especially important, when the authors want to claim that the shift of TM6 towards TM3 observed in their structures requires a small side chain at this position. However, it would be reasonable for me, if the authors would just rephrase their discussion by highlighting the stabilizing effect of the H-bond formed by S234 without discussing the side chain size at this position.

lines 251-262: This part requires a short introduction. In the current form, it is totally disconnected from the rest. What do the authors want to say and why? To my understanding, the authors compared different GPCR-Gi complexes and want to state that the relative orientation between the G protein and the receptors is different (as judged by the position of the alphaN helix).

Lines 326-328: This is not clear to me. Why is the unique ligand binding pocket necessary to bind the ligand at concentrations higher than physiological concentrations? This requires some additional explanations.

Line 351: Please, rephrase this sentence. The authors propose, that: "these findings indicate that the inactive conformation of MrgD is not stable and can therefore assume the active conformation more quickly, resulting in higher basal activity." Considering the energy landscapes of GPCRs, I would rather say that the absent sodium binding site and the modified DRY, PIF and CWxP motifs shift the conformational equilibrium of the receptor from the inactive states to the active state population. This could be due to the potential lower energy of the active state in comparison to other class A GPCRs and a higher energy state of the inactive conformations. Another possibility is that the energy barrier between inactive states and active state populations is reduced. This would allow for faster transitions between the inactive and active states, and vice versa. Based on the data presented in this paper, it is hard to distinguish between these two options. Therefore, I would just propose to mention that the modifications might destabilize the inactive state, which shifts the conformational equilibrium more toward the active state of the receptor.

Minor:

line 184: Please, add the article in front of "canonical"

line 192: Please complete the sentence: The most characteristic "conformational change" ...

line 194: Please, complete the sentence: in good agreement with other "active state" GPCR "structures".

line 198: please, change has to "forms"

line 225: "...mutants abolished." – Please, add information about what was abolished.

Line 265: replace "has" with "shows"

Lines 276-278: please, rephrase the sentence. I assume you wanted to say: We first measured the basal activity of point mutants of residues within the ligand-binding site, which are involved in the interaction between TM3 and TM6.

line 293: please, change "or" to "and/or"

line 307: please, change "has" to "have"

line 312: please, add "being" before restraint

line 317-318: please, change to "...as a necessary adaptation to form a ligand-binding pocket, allow proper folding, and enable receptor activation."

line 320: please, change "kings" to "kink"

line 339: please, add: "similar" before binding position

line 348: please, add an a in front of Q

line 349: please, change to "seen in the inactive state of other GPCRs"

line 350: please, change "mutants" to "mutations"

line 355: please, change "among" to "between"

line 369: please, change "indicating the diversity of ligands..." to "providing an explanation for the ligand binding specificity of different MRGPR receptors"

lines 373-374: "The structure of MRGPR family proteins in the inactive state has yet been reported so far." Please, change to: "The structure of MRGPR family proteins in the inactive state has not been reported so far" or "No inactive state structures of MRGPR family proteins have not been reported yet".

Reviewer #4 (Remarks to the Author):

Page2, line 63

I believe "RAD pathway" should be renin-angiotensin system (RAS) pathway for clarity.

Page7, line 225-226

These two "activity" should be efficacy and potency, respectively based on the Fig4C.

Other than these minor comments, I think the authors addressed all the questions.

Response to Reviews:

Specific answers to Reviewer #1 comments

Reviewer #1 (Remarks to the Author):

The authors have modified the introduction portion significantly. The author performed more cryo-EM data processing, which improves the cryo-EM structures. Authors modified several figures and represented the figures more appropriately, which will be easier for the general reader. Additionally, the authors explained most of the Reviewer's suggestions. The authors addressed most of the questions, and their responses are convincing. Therefore, in my opinion, this manuscript would be accepted for publication.

Specific answers to Reviewer #2 comments

Reviewer #2 (Remarks to the Author):

In this revised manuscript by Suzuki and colleagues, the authors describe structures of MrgD in b-alanine and ligand-free states. The structures presented with the revised manuscript are of higher quality and provide a structural framework for investigating MrgD and related MRGPR family receptors. In response to reviewer comments, the authors substantially revised their interpretations of the structures. Despite the changes, several of the claims made by the authors are only weakly supported and additional changes to the text will be necessary prior to publication.

Authors:

We thank Reviewer #2 for the constructive and helpful comments to improve our manuscript.

Comments:

1. In the abstract, the authors claim that their structures provide a structural basis for the high basal activity. While the authors do determine structures in ligand-bound and ligand-free activated states, the mechanism of ligand-independent activation is still not clear. The authors show that mutation of Y109 and S234 reduces ligand-independent activation. However, they show in figure 4 that these mutations also diminish ligand-dependent activation and thus it is unclear if these mutations simply break activation or are specific to basal activity.

Authors:

The phrase "structural basis for the high basal activity" has been removed from the abstract and discussion in the revised manuscript because the ligand-independent activation mechanism has not been fully clarified.

We respectfully say that we do not intend to describe that Y109 and S243 contribute specifically to basal activity. We believe that ligand-dependent activity and basal activity should be discussed separately, as has been the case in the recent studies (Wang et al. *Nat Commun* (2021), Zhang et al. *Nat Commun* (2021), Qin et al. *Nat Commun* (2022)). Our results indicate that the high basal activity of MrgD is not due to the stabilization of the activated state by the interaction of specific amino acids in the extracellular region. For example, R103A and D179A have eliminated the Gi signaling activity, but the basal activity was comparable to WT. The basal activity is reduced for Y109A, Y109F, and S234A, but the Gi signaling activity is retained. Therefore, these amino acids, Y109 and S234, are at least involved in the high basal activity of MrgD. Nevertheless, as pointed out, this does not rule out the possibility that these mutations break activation itself. We avoided "essential" and changed it to "involved" in the result section.

Page 8, line 274

Measurement of the basal activity of the mutants of these residues revealed that the Y109A, Y106F, and S234A mutations reduced the basal activity (Fig. 6d), suggesting that these two hydrogen bonds are involved in the basal activity of MrgD.

2. The ligand-bound MD simulations were performed starting from the pose assigned based on the local chemical environment of the ligand as the density was insufficient for unambiguous modeling. Did the authors also attempt any alternative poses to determine if their approach could “rigorously evaluate the binding orientation of b-alanine”? Especially as the affinity of MrgD for b-alanine is quite low.

Authors:

We admit that the original map quality for modeling β -alanine was not sufficient to unambiguously determine the orientation. We performed additional MD simulations. When we generated the model that has β -alanine in the opposite orientation and started the simulation in three independent runs, the orientation of β -alanine was reversed in the equilibration steps in all runs and did not change after that (response Fig.1). These results suggest that our modeling of β -alanine is reasonable.

However, the word “rigorously” in the text is an overstatement, so it has been removed.

Response Figure 1. MD simulation to determine β -alanine binding orientation

Superposition of the start model (grey) in which β -alanine is oriented oppositely and the last snapshots from the three independent 250 ns MD simulations (run1: cyan, run2: blue, run3: slate blue). The snapshots show that β -alanine orientation is reversed in all simulations.

3. The MD analysis indicates that the activated conformation is less stable for the apo state. However, MrgD has basal activity. How do the dynamics of TM6 compare to receptors with low basal activity?

Authors:

Honestly, we cannot adequately address this question. We think TM6 dynamics of receptors with low basal activity would be stable in a closed-form because the receptors have conserved motifs such as sodium ion binding-site and ion-lock between TM3 and TM6, stabilizing the inactive state when the ligand is not bound. This has been implicated by Wifling et al. (Chem. Eur. J. 2019, 25, 14613 – 14624).

In this study, since the MrgD structure has not been solved for its apo inactive conformation without Gi or ligand, the dynamics of TM6 in the other GPCRs cannot be compared. We cannot demonstrate the mechanistic insights of the basal activity from our MD simulation, and we have not described basal activity associated with MD in the manuscript.

Because Reviewer #3 kindly suggested the supportive interpretation of the conformational equilibrium toward the activated state, we added the following sentence in the Discussion section.

Page 10, line 344

These findings indicate that the absent sodium binding site and some of the conserved motif's modifications might destabilize the inactive state and shift the conformational equilibrium of the receptor from the inactive state to the active state population.

Minor points:

1. Introduction section between Line 46-60 is a little repetitive. They mention b-alanine activating Gi, Gq, and Gs activation at multiple nonsequential points. This section could be refined.

Authors:

As suggested, the sentence of “Thus, β -alanine can induce the Gs, Gq, or Gi pathway through activation of MrgD” has been removed.

2. The introduction has been modified to shift focus even further to the apo conformation and its role in basal activity, but as discussed above, it is not clear what mechanistic insight can be drawn from their findings.

Authors:

Because the mutations in amino acids around the extracellular ligand-binding pocket did not show significant differences in the basal activity (supplementary Fig. 12), we only show that the two hydrogen bonds mediated by Y109 and S234 may be involved in the basal activity of MrgD. We think that the presence of G protein is required for MrgD to stabilize the activated state in the absence of ligand. An inverse agonist bound structure would provide a more detailed mechanism, but no inverse agonist is currently available.

Between Line 102-107 they describe how they co-expressed and co-purified the complex using b-alanine but provide no information about how they generated the apo complex — it should be mentioned here as well (as discussed in M&M Line 644).

Authors:

The following text has been added.

Page 3, line 102

In the case of the apo state, no ligand was added during all the purification steps.

Specific answers to Reviewer #3 comments

Reviewer #3 (Remarks to the Author):

The authors sufficiently addressed my previous concerns. The manuscript has been significantly improved by the addition of new mutational data, MD simulations and updated figures. However, I still have some comments regarding the role of S234 and the reason for the basal activity of the receptor. In the following I list some major and minor points the author should address before I can recommend publication of their manuscript:

Authors:

We thank Reviewer #3 for the constructive and helpful comments to improve our manuscript.

Major comment;

lines 215-219: I am still not sure what the authors want to say here. The S234A mutation proves that the H-bond between this side chain and residue S268 is important for the stabilization of the active state, which has been now nicely addressed by the authors on Page9 line 279. However, in order to investigate the impact of the side chain size at this position on the reorientation of F230, the authors should also substitute the side chain with larger ones (e.g. Tyr or the canonical Trp). This is especially important, when the authors want to claim that the shift of TM6 towards TM3 observed in their structures requires a small side chain at this position. However, it would be reasonable for me, if the authors would just rephrase their discussion by highlighting the stabilizing effect of the H-bond formed by S234 without discussing the side chain size at this position.

Authors:

The description of the side chain size of S234 was removed from the results section because the evidence is weak and hardly contributes to the debate in our manuscript. Specifically, “To investigate whether the small side chain of S234^{6.48} can directly affect the rotamer of F230^{6.44},” has been eliminated.

lines 251-262: This part requires a short introduction. In the current form, it is totally disconnected from the rest. What do the authors want to say and why? To my understanding, the authors compared different GPCR-Gi complexes and want to state that the relative orientation between the G protein and the receptors is different (as judged by the position of the alphaN helix).

Authors:

We would like to mention that the relative orientation of the N-terminus is different. Therefore, we have removed the roundabout sentences and revised the paragraph as follows.

Page 8, line 246

The MrgD-Gi complex is stabilized by hydrogen bonds contributed by the backbone carbonyls of A31, R137^{34,57}, and C135^{34,55} and the side chains of K134^{34,54} and R32 (Fig. 5d, Supplementary Fig. 11a). However, The interactions between ICL2 and Gi are not common in other Gi-bound GPCR complexes (Supplementary Fig. 11b-g). When the receptor portion of each complex is superimposed, the orientation of the α N helix of Gi is diverse (Supplementary Fig. 11h). The different interactions between ICL2 and Gi may be one of the factors that contribute to the large displacement of α N helix in Gi-bound GPCRs.

Lines 326-328: This is not clear to me. Why is the unique ligand binding pocket necessary to bind the ligand at concentrations higher than physiological concentrations? This requires some additional explanations.

Authors:

As suggested, we cannot claim whether the shallow ligand-binding pocket of MrgD is essential for the recognition of β -alanine in the physiological milieu from this study. We changed the interpretation in the discussion section as follows.

Page 10, line 319

The shallow and solvent-exposed ligand-binding pocket may contribute to the low ligand-binding potency of β -alanine.

Line 351: Please, rephrase this sentence. The authors propose, that: “;these findings indicate that the inactive conformation of MrgD is not stable and can therefore assume the active conformation more quickly, resulting in higher basal activity.”; Considering the energy landscapes of GPCRs, I would rather say that the absent sodium binding site and the modified DRY, PIF and CWxP motifs shift the conformational equilibrium of the receptor from the inactive states to the active state population. This could be due to the potential lower energy of the active state in comparison to other class A GPCRs and a higher energy state of the inactive conformations. Another possibility is that the energy barrier between inactive states and active state populations is reduced. This would allow for faster transitions between the inactive and active states, and vice versa. Based on the data presented in this paper, it is hard to distinguish

between these two options. Therefore, I would just propose to mention that the modifications might destabilize the inactive state, which shifts the conformational equilibrium more toward the active state of the receptor.

Authors:

Thank you for pointing this out. We agree that the Reviewer's comments as a very rational interpretation. We have revised the text as follows.

Page 10, line 344

These findings indicate that the absent sodium binding site and the modifications of the conserved motifs might destabilize the inactive state and shift the conformational equilibrium of the receptor from the inactive state to the active state population.

Minor comments:

line 184: Please, add the article in front of "canonical"

Authors:

We modified the text.

line 192: Please complete the sentence: The most characteristic "conformational change" ...

Authors:

We modified the text.

line 194: Please, complete the sentence: in good agreement with other "active state" GPCR "structures".

Authors:

We modified the text.

line 198: please, change has to "forms"

Authors:

We modified the text.

line 225: "...mutants abolished." Please, add information about what was abolished.

Authors:

We modified the text.

Line 265: replace "has"; with "shows"

Authors:

We modified the text.

Lines 276-278: please, rephrase the sentence. I assume you wanted to say: We first measured the basal activity of point mutants of residues within the ligand-binding site, which are involved in the interaction between TM3 and TM6.

Authors:

We modified the text.

line 293: please, change "or" to "and/or"

Authors:

We modified the text.

line 307: please, change "has" to "have"

Authors:

We modified the text.

line 312: please, add “being” before restraint

Authors:

We modified the text.

line 317-318: please, change to “..as a necessary adaptation to form a ligand-binding pocket, allow proper folding, and enable receptor activation.”

Authors:

We modified the text.

line 320: please, change "kings" to "kink"

Authors:

We modified the text.

line 339: please, add: “similar” before binding position

Authors:

We modified the text.

line 348: please, add an a in front of Q

Authors:

We modified the text.

line 349: please, change to "seen in the inactive state of other GPCRs"

Authors:

We modified the text.

line 350: please, change "mutants" to "mutations"

Authors:

We modified the text.

line 355: please, change "among" to "between"

Authors:

We modified the text.

line 369: please, change "indicating the diversity of ligands...;" to "providing an explanation for the ligand binding specificity of different MRGPR receptors"

Authors:

We modified the text.

lines 373-374: "The structure of MRGPR family proteins in the inactive state has yet been reported so far."; Please, change to: "The structure of MRGPR family proteins in the inactive state has not been reported so far" or "No inactive state structures of MRGPR family proteins have not been reported yet".

Authors:

We modified the text.

Specific answers to Reviewer #4 comments

Reviewer #4 (Remarks to the Author):

Comment;

Page2, line 63

I believe "RAD pathway" should be renin-angiotensin system (RAS) pathway for clarity.

Authors:

We modified the text.

Page7, line 225-226

These two “activity” should be efficacy and potency, respectively based on the Fig4C.

Authors:

We modified the text.

Other than these minor comments, I think the authors addressed all the questions.

Authors:

We appreciate this comment.

REVIEWERS' COMMENTS:

Reviewer #2 (Remarks to the Author):

The authors have sufficiently addressed my concerns and the manuscript is suitable for publication.

Reviewer #3 (Remarks to the Author):

The authors significantly improved the manuscript and I can now recommend to accept it for publication after these final minor edits have been fixed:

Line 129: I assume that the authors mean that the amino group of beta-alanine is sufficiently close to form a H-bond with the backbone of W241 and the carboxyl group of the ligand with the backbone of C164.

Line 340: The Na⁺-binding site is important for stabilizing the inactive state of some GPCRs, but others do not show this allosteric ion binding site. I would rather say that it is important for the stabilization of some GPCRs.

Line 343: Please, provide the references for the structures with an intact ionic lock.

Line 346: I would use plural form "populations", because GPCRs have been shown to sample multiple inactive and active states (... from inactive state to active state populations.)